# Implicit MLE: Backpropagating Through Discrete Exponential Family Distributions

**Mathias Niepert**
NEC Laboratories Europe
mathias.niepert@neclab.eu

**Pasquale Minervini**
University College London
p.minervini@ucl.ac.uk

**Luca Franceschi**
Istituto Italiano di Tecnologia
University College London
ucablfr@ucl.ac.uk

## Abstract

Combining discrete probability distributions and combinatorial optimization problems with neural network components has numerous applications but poses several challenges. We propose Implicit Maximum Likelihood Estimation (I-MLE), a framework for end-to-end learning of models combining discrete exponential family distributions and differentiable neural components. I-MLE is widely applicable as it only requires the ability to compute the most probable states and does not rely on smooth relaxations. The framework encompasses several approaches such as perturbation-based implicit differentiation and recent methods to differentiate through black-box combinatorial solvers. We introduce a novel class of noise distributions for approximating marginals via perturb-and-MAP. Moreover, we show that I-MLE simplifies to maximum likelihood estimation when used in some recently studied learning settings that involve combinatorial solvers. Experiments on several datasets suggest that I-MLE is competitive with and often outperforms existing approaches which rely on problem-specific relaxations.

## 1 Introduction

While deep neural networks excel at perceptual tasks, they tend to generalize poorly whenever the problem at hand requires some level of symbolic manipulation or reasoning, or exhibit some (known) algorithmic structure. Logic, relations, and explanations, as well as decision processes, frequently find natural abstractions in discrete structures, ill-captured by the continuous mappings of standard neural nets. Several application domains, ranging from relational and explainable ML to discrete decision-making [Mišić and Perakis, 2020], could benefit from general-purpose learning algorithms whose inductive biases are more amenable to integrating symbolic and neural computation. Motivated by these considerations, there is a growing interest in end-to-end learnable models incorporating discrete components that allow, e.g., to sample from discrete latent distributions [Jang et al., 2017, Paulus et al., 2020] or solve combinatorial optimization problems [Pogančić et al., 2019, Mandi et al., 2020]. Discrete energy-based models (EBMs) [LeCun et al., 2006] and discrete world models [Hafner et al., 2020] are additional examples of neural network based models that require the ability to backpropagate through discrete probability distributions.

For complex discrete distributions, it is intractable to compute the exact gradients of the expected loss. For combinatorial optimization problems, the loss is discontinuous, and the gradients are zero almost everywhere. The standard approach revolves around problem-specific smooth relaxations, which allow one to fall back to (stochastic) backpropagation. These strategies, however, require tailor-made relaxations, presuppose access to the constraints and are, therefore, not always feasible nor tractable for large state spaces. Moreover, reverting to discrete outputs at test time may cause unexpected behavior. In other situations, discrete outputs are required at training time because one has to make one of a number of discrete choices, such as accessing discrete memory or deciding on an action in a game.

35th Conference on Neural Information Processing Systems (NeurIPS 2021).

With this paper, we take a step towards the vision of general-purpose algorithms for hybrid learning systems. Specifically, we consider settings where the discrete component(s), embedded in a larger computational graph, are discrete random variables from the constrained exponential family[1]. Grounded in concepts from Maximum Likelihood Estimation (MLE) and perturbation-based implicit differentiation, we propose Implicit Maximum Likelihood Estimation (I-MLE). To approximate the gradients of the discrete distributions' parameters, I-MLE computes, at each update step, a *target distribution* $q$ that depends on the loss incurred from the discrete output in the forward pass. In the backward pass, we approximate maximum likelihood gradients by treating $q$ as the empirical distribution. We propose ways to derive target distributions and introduce a novel family of noise perturbations well-suited for approximating marginals via perturb-and-MAP. I-MLE is general-purpose as it only requires the ability to compute most probable states and not faithful samples or probabilistic inference. In summary, we make the following contributions:

1. We propose implicit maximum likelihood estimation (I-MLE) as a framework for computing gradients with respect to the parameters of discrete exponential family distributions;

2. We show that this framework is useful for backpropagating gradients through *both* discrete probability distributions and discrete combinatorial optimization problems;

3. I-MLE requires two ingredients: a family of target distribution $q$ and a method to sample from complex discrete distributions. We propose two families of target distributions and a family of noise-distributions for Gumbel-max (perturb-and-MAP) based sampling.

4. We show that I-MLE simplifies to *explicit* maximum-likelihood learning when used in some recently studied learning settings involving combinatorial optimization solvers.

5. Extensive experimental results suggest that I-MLE is flexible and competitive compared to the straight-through and relaxation-based estimators.

Instances of the I-MLE framework can be easily integrated into modern deep learning pipelines, allowing one to readily utilize several types of discrete layers with minimal effort. We provide implementations and Python notebooks at `https://github.com/nec-research/tf-imle`

## 2 Problem Statement and Motivation

We consider models described by the equations

$$\boldsymbol{\theta} = h_{\boldsymbol{v}}(\boldsymbol{x}), \quad \boldsymbol{z} \sim p(\boldsymbol{z}; \boldsymbol{\theta}), \quad \boldsymbol{y} = f_{\boldsymbol{u}}(\boldsymbol{z}), \quad (1)$$

where $\boldsymbol{x} \in \mathcal{X}$ and $\boldsymbol{y} \in \mathcal{Y}$ denote feature inputs and target outputs, $h_{\boldsymbol{v}} : \mathcal{X} \to \Theta$ and $f_{\boldsymbol{u}} : \mathcal{Z} \to \mathcal{Y}$ are smooth parameterized maps, and $p(\boldsymbol{z}; \boldsymbol{\theta})$ is a discrete probability distribution.

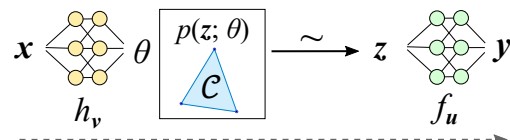

Figure 1: Illustration of the addressed learning problem. $\boldsymbol{z}$ is the discrete (latent) structure.

Given a set of examples $\mathcal{D} = \{(\hat{\boldsymbol{x}}_j, \hat{\boldsymbol{y}}_j)\}_{j=1}^{N}$, we are concerned with learning the parameters $\boldsymbol{\omega} = (\boldsymbol{v}, \boldsymbol{u})$ of (1) by finding approximate solutions of $\min_{\boldsymbol{\omega}} \sum_j L(\hat{\boldsymbol{x}}_j, \hat{\boldsymbol{y}}_j; \boldsymbol{\omega})/N$. The training error $L$ is typically defined as:

$$L(\hat{\boldsymbol{x}}, \hat{\boldsymbol{y}}; \boldsymbol{\omega}) = \mathbb{E}_{\hat{\boldsymbol{z}} \sim p(\boldsymbol{z}; \hat{\boldsymbol{\theta}})} [\ell(f_{\boldsymbol{u}}(\hat{\boldsymbol{z}}), \hat{\boldsymbol{y}})] \quad \text{with} \quad \hat{\boldsymbol{\theta}} = h_{\boldsymbol{v}}(\hat{\boldsymbol{x}}), \quad (2)$$

where $\ell : \mathcal{Y} \times \mathcal{Y} \to \mathbb{R}^{+}$ is a point-wise loss function. Fig. 1 illustrates the setting. For example, an interesting instance of (1) and (2) arises in *learning to explain* user reviews [Chen et al., 2018] where the task is to infer a target sentiment score (e.g. w.r.t. the quality of a product) from a review while also providing a concise explanation of the predicted score by selecting a subset of exactly $k$ words (cf. Example 2). In Section 6, we present experiments precisely in this setting. As anticipated in the introduction, we restrict the discussion to instances in which $p(\boldsymbol{z}; \boldsymbol{\theta})$ belongs to the (constrained) *discrete exponential family*, which we now formally introduce.

Let $\boldsymbol{Z}$ be a vector of discrete random variables over a state space $\mathcal{Z}$ and let $\mathcal{C} \subseteq \mathcal{Z}$ be the set of states that satisfy a given set of *linear constraints*.[2] Let $\boldsymbol{\theta} \in \Theta \subseteq \mathbb{R}^m$ be a real-valued parameter vector.

---

[1]This includes integer linear programs via a natural link that we outline in Example 3.

[2]For the sake of simplicity we assume $\mathcal{Z} \subseteq \{0, 1\}^m$. The set $\mathcal{C}$ is the integral polytope spanned by the given, problem-specific, linear constraints.

The probability mass function (PMF) of a discrete constrained exponential family r.v. is:

$$p(\boldsymbol{z};\boldsymbol{\theta}) = \begin{cases} \exp\left(\langle \boldsymbol{z}, \boldsymbol{\theta} \rangle / \tau - A(\boldsymbol{\theta})\right) & \text{if } \boldsymbol{z} \in \mathcal{C}, \\ 0 & \text{otherwise.} \end{cases} \tag{3}$$

Here, $\langle \cdot, \cdot \rangle$ is the inner product and $\tau$ the temperature, which, if not mentioned otherwise, is assumed to be 1. $A(\boldsymbol{\theta})$ is the log-partition function defined as $A(\boldsymbol{\theta}) = \log\left(\sum_{\boldsymbol{z} \in \mathcal{C}} \exp\left(\langle \boldsymbol{z}, \boldsymbol{\theta} \rangle / \tau\right)\right)$. We call $\langle \boldsymbol{z}, \boldsymbol{\theta} \rangle$ the *weight* of the state $\boldsymbol{z}$. The *marginals* (expected value, mean) of the r.v.s $\mathbf{Z}$ are defined as $\boldsymbol{\mu}(\boldsymbol{\theta}) := \mathbb{E}_{\hat{\boldsymbol{z}} \sim p(\boldsymbol{z};\boldsymbol{\theta})}[\hat{\boldsymbol{z}}]$. Finally, the most probable or Maximum A-Posteriori (MAP) states are defined as $\texttt{MAP}(\boldsymbol{\theta}) := \arg\max_{\boldsymbol{z} \in \mathcal{C}} \langle \boldsymbol{z}, \boldsymbol{\theta} \rangle$. The family of probability distributions we define here captures a broad range of settings and subsumes probability distributions such as positive Markov random fields and statistical relational formalisms [Wainwright and Jordan, 2008, Raedt et al., 2016]. We now discuss some examples which we will use in the experiments. Crucially, in Example 3 we establish the link between the constrained exponential family and integer linear programming (ILP) identifying the ILP cost coefficients with the distribution's parameters $\boldsymbol{\theta}$.

**Example 1** (Categorical Variables). *An $m$-way (one-hot) categorical variable corresponds to $p(\boldsymbol{z};\boldsymbol{\theta}) = \exp\left(\langle \boldsymbol{z}, \boldsymbol{\theta} \rangle - A(\boldsymbol{\theta})\right)$, subject to the constraint $\langle \boldsymbol{z}, \mathbf{1} \rangle = 1$, where $\mathbf{1}$ is a vector of ones.*

As $\mathcal{C} = \{\mathbf{e}_i\}_{i=1}^{m}$, where $\mathbf{e}_i$ is the $i$-th vector of the canonical base, the parameters of the above distribution coincide with the weights, which are often called *logits* in this context. The marginals $\boldsymbol{\mu}$ coincide with the PMF and can be expressed through a closed-form smooth function of $\boldsymbol{\theta}$: the softmax. This facilitates a natural relaxation that involves using $\boldsymbol{\mu}(\boldsymbol{\theta})$ in place of $\boldsymbol{z}$ [Jang et al., 2017]. The convenient properties of the categorical distribution, however, quickly disappear even for slightly more complex distributions, as the following example shows.

**Example 2** ($k$-subset Selection). *Assume we want to sample binary $m$-dimensional vectors with $k$ ones. This amounts to replacing the constraint in Example 1 by the constraint $\langle \boldsymbol{z}, \mathbf{1} \rangle = k$.*

Here, a closed-form expression for the marginals does not exist: sampling from this distribution requires computing the $\binom{m}{k} = O(m^k)$ weights (if $k \leq m/2$). Computing MAP states instead takes time linear in $m$.

**Example 3** (Integer Linear Programs). *Consider the combinatorial optimization problem given by the integer linear program $\arg\min_{\boldsymbol{z} \in \mathcal{C}} \langle \boldsymbol{z}, \mathbf{c} \rangle$, where $\mathcal{C}$ is an integral polytope and $\mathbf{c} \in \mathbb{R}^m$ is a vector of cost coefficients, and let $\boldsymbol{z}^*(\mathbf{c})$ be the set of its solutions. We can associate to the ILP the family (indexed by $\tau > 0$) of probability distributions $p(\boldsymbol{z};\boldsymbol{\theta})$ from (3), with $\mathcal{C}$ the ILP polytope and $\boldsymbol{\theta} = -\mathbf{c}$. Then, for every $\tau > 0$, the solutions of the ILP correspond to the MAP states: $\texttt{MAP}(\boldsymbol{\theta}) = \arg\max_{\boldsymbol{z} \in \mathcal{C}} \langle \boldsymbol{z}, \boldsymbol{\theta} \rangle = \boldsymbol{z}^*(\mathbf{c})$ and for $\tau \to 0$ one has that $\Pr(\mathbf{Z} \in \boldsymbol{z}^*(\mathbf{c})) \to 1$.*

Many problems of practical interest can be expressed as ILPs, such as finding shortest paths, planning and scheduling problems, and inference in propositional logic.

## 3 The Implicit Maximum Likelihood Estimator

In this section, we develop and motivate a family of general-purpose gradient estimators for Eq. (2) that respect the structure of $\mathcal{C}$ . [3] Let $(\hat{\boldsymbol{x}}, \hat{\boldsymbol{y}}) \in \mathcal{D}$ be a training example and $\hat{\boldsymbol{z}} \sim p(\boldsymbol{z}; h_{\boldsymbol{v}}(\hat{\boldsymbol{x}}))$. The gradient of $L$ w.r.t. $\boldsymbol{u}$ is given by $\nabla_{\boldsymbol{u}} L(\hat{\boldsymbol{x}}, \hat{\boldsymbol{y}}; \boldsymbol{\omega}) = \mathbb{E}_{\hat{\boldsymbol{z}}}[\partial_{\boldsymbol{u}} f_{\boldsymbol{u}}(\hat{\boldsymbol{z}})^{\intercal} \nabla_{\boldsymbol{y}} \ell(\boldsymbol{y}, \hat{\boldsymbol{y}})]$ with $\boldsymbol{y} = f_{\boldsymbol{u}}(\hat{\boldsymbol{z}})$, which may be estimated by drawing one or more samples from $p$. Regarding $\nabla_{\boldsymbol{v}} L$, one has

$$\nabla_{\boldsymbol{v}} L(\hat{\boldsymbol{x}}, \hat{\boldsymbol{y}}; \boldsymbol{\omega}) = \partial_{\boldsymbol{v}} h_{\boldsymbol{v}}(\hat{\boldsymbol{x}})^{\intercal} \nabla_{\boldsymbol{\theta}} L(\hat{\boldsymbol{x}}, \hat{\boldsymbol{y}}; \boldsymbol{\omega}), \tag{4}$$

where the major challenge is to compute $\nabla_{\boldsymbol{\theta}} L$. A standard approach is to employ the score function estimator (SFE) which typically suffers from high variance. Whenever a pathwise derivative estimator (PDE) is available it is usually the preferred choice [Schulman et al., 2015]. In our setting, however, the PDE is not readily applicable since $\boldsymbol{z}$ is discrete and, therefore, every (exact) reparameterization path would be discontinuous. Various authors developed (biased) adaptations of the PDE for discrete r.v.s (see Section 5). These involve either smooth approximations of $p(\boldsymbol{z};\boldsymbol{\theta})$ or approximations of the derivative of the reparameterization map. *Our proposal departs from these two routes and instead involves the formulation of an implicit maximum likelihood estimation problem.* In a nutshell, I-MLE

---

[3]The derivations are adaptable to other types of losses defined over the outputs of Eq. (1).

**Algorithm 1** Instance of I-MLE with perturbation-based implicit differentiation.

**function** FORWARDPASS($\boldsymbol{\theta}$)
    *// Sample from the noise distribution $\rho(\boldsymbol{\epsilon})$*
    $\boldsymbol{\epsilon} \sim \rho(\boldsymbol{\epsilon})$
    *// Compute a MAP state of perturbed $\boldsymbol{\theta}$*
    $\hat{\boldsymbol{z}} = \texttt{MAP}(\boldsymbol{\theta} + \boldsymbol{\epsilon})$
    **save** $\boldsymbol{\theta}, \boldsymbol{\epsilon}$, and $\hat{\boldsymbol{z}}$ for the backward pass
    **return** $\hat{\boldsymbol{z}}$

**function** BACKWARDPASS($\nabla_{\boldsymbol{z}}\ell(f_{\boldsymbol{u}}(\boldsymbol{z}), \hat{\boldsymbol{y}}), \lambda$)
    **load** $\boldsymbol{\theta}, \boldsymbol{\epsilon}$, and $\hat{\boldsymbol{z}}$ from the forward pass
    *// Compute target distribution parameters*
    $\boldsymbol{\theta}' = \boldsymbol{\theta} - \lambda\nabla_{\boldsymbol{z}}\ell(f_{\boldsymbol{u}}(\boldsymbol{z}), \hat{\boldsymbol{y}})$
    *// Single sample I-MLE gradient estimate*
    $\widehat{\nabla}_{\boldsymbol{\theta}}\mathcal{L}(\hat{\boldsymbol{\theta}}, \hat{\boldsymbol{\theta}}') = \hat{\boldsymbol{z}} - \texttt{MAP}(\boldsymbol{\theta}' + \boldsymbol{\epsilon})$
    **return** $\widehat{\nabla}_{\boldsymbol{\theta}}\mathcal{L}(\hat{\boldsymbol{\theta}}, \hat{\boldsymbol{\theta}}')$

is a (biased) estimator that replaces $\nabla_{\boldsymbol{\theta}}L$ in Eq. (4) with $\widehat{\nabla}_{\boldsymbol{\theta}}\mathcal{L}$, where $\mathcal{L}$ is an implicitly defined MLE objective and $\widehat{\nabla}$ is an estimator of the gradient.

We now focus on deriving the (implicit) MLE objective $\mathcal{L}$. Let us assume we can, for any given $\hat{\boldsymbol{y}}$, construct an exponential family distribution $q(\boldsymbol{z}; \boldsymbol{\theta}')$ that, ideally, is such that

$$\mathbb{E}_{\hat{\boldsymbol{z}} \sim q(\boldsymbol{z};\boldsymbol{\theta}')}[\ell(f_{\boldsymbol{u}}(\hat{\boldsymbol{z}}), \hat{\boldsymbol{y}}] \leq \mathbb{E}_{\hat{\boldsymbol{z}} \sim p(\boldsymbol{z};\boldsymbol{\theta})}[\ell(f_{\boldsymbol{u}}(\hat{\boldsymbol{z}}), \hat{\boldsymbol{y}})]. \tag{5}$$

We will call $q$ the *target distribution*. The idea is that, by making $p$ more similar to $q$ we can (iteratively) reduce the model loss $L(\hat{\boldsymbol{x}}, \hat{\boldsymbol{y}}; \boldsymbol{\omega})$. To this purpose, we define $\mathcal{L}$ as the MLE objective[4] between the model distribution $p$ with parameters $\boldsymbol{\theta}$ and the target distribution $q$ with parameters $\boldsymbol{\theta}'$:

$$\mathcal{L}(\boldsymbol{\theta}, \boldsymbol{\theta}') \coloneqq -\mathbb{E}_{\hat{\boldsymbol{z}} \sim q(\boldsymbol{z};\boldsymbol{\theta}')}[\log p(\hat{\boldsymbol{z}}; \boldsymbol{\theta})] = \mathbb{E}_{\hat{\boldsymbol{z}} \sim q(\boldsymbol{z};\boldsymbol{\theta}')}[A(\boldsymbol{\theta}) - \langle\hat{\boldsymbol{z}}, \boldsymbol{\theta}\rangle] \tag{6}$$

Now, exploiting the fact that $\nabla_{\boldsymbol{\theta}}A(\boldsymbol{\theta}) = \boldsymbol{\mu}(\boldsymbol{\theta})$, we can compute the gradient of $\mathcal{L}$ as

$$\nabla_{\boldsymbol{\theta}}\mathcal{L}(\boldsymbol{\theta}, \boldsymbol{\theta}') = \boldsymbol{\mu}(\boldsymbol{\theta}) - \mathbb{E}_{\hat{\boldsymbol{z}} \sim q(\boldsymbol{z};\boldsymbol{\theta}')}[\hat{\boldsymbol{z}}] = \boldsymbol{\mu}(\boldsymbol{\theta}) - \boldsymbol{\mu}(\boldsymbol{\theta}'), \tag{7}$$

that is , the difference between the marginals of the current distribution $p$ and the marginals of the target distribution $q$, also equivalent to the gradient of the KL divergence between $p$ and $q$.

We will not use Eq. (7) directly, as computing the marginals is, in general, a #P-hard problem and scales poorly with the dimensionality $m$. MAP states are typically less expensive to compute (e.g. see Example 2) and are often used directly to approximate $\boldsymbol{\mu}(\boldsymbol{\theta})$[5] or to compute perturb-and-MAP approximations, where $\boldsymbol{\mu}(\boldsymbol{\theta}) \approx \mathbb{E}_{\boldsymbol{\epsilon}\sim\rho(\boldsymbol{\epsilon})}\texttt{MAP}(\boldsymbol{\theta} + \boldsymbol{\epsilon})$ where $\boldsymbol{\epsilon} \sim \rho(\boldsymbol{\epsilon})$ is an appropriate *noise distribution* with domain $\mathbb{R}^m$. In this work we follow – and explore in more detail in Section 3.2 – the latter approach (also referred to as the Gumbel-max trick [cf. Papandreou and Yuille, 2011]), a strategy that retains most of the computational advantages of the pure MAP approximation but may be less crude. Henceforth, we only assume access to an algorithm to compute MAP states (such as a standard ILP solver in the case of Example 3) and rephrase Eq. (1) as

$$\boldsymbol{\theta} = h_{\boldsymbol{v}}(\boldsymbol{x}), \quad \boldsymbol{z} = \texttt{MAP}(\boldsymbol{\theta} + \boldsymbol{\epsilon}) \text{ with } \boldsymbol{\epsilon} \sim p(\boldsymbol{\epsilon}), \quad \boldsymbol{y} = f_{\boldsymbol{u}}(\boldsymbol{z}). \tag{8}$$

With Eq. (8) in place, the general expression for the I-MLE estimator is $\widehat{\nabla}_{\boldsymbol{v}}L(\boldsymbol{x}, \boldsymbol{y}; \boldsymbol{\omega}) = \partial_{\boldsymbol{v}}h_{\boldsymbol{v}}(\hat{\boldsymbol{x}})^{\mathsf{T}}\widehat{\nabla}_{\boldsymbol{\theta}}\mathcal{L}(\boldsymbol{\theta}, \boldsymbol{\theta}')$ with $\boldsymbol{\theta} = h_{\boldsymbol{v}}(\hat{\boldsymbol{x}})$ where, for $S \in \mathbb{N}^+$:

$$\widehat{\nabla}_{\boldsymbol{\theta}}\mathcal{L}(\boldsymbol{\theta}, \boldsymbol{\theta}') = \frac{1}{S}\sum_{i=1}^{S}[\texttt{MAP}(\boldsymbol{\theta} + \boldsymbol{\epsilon}_i) - \texttt{MAP}(\boldsymbol{\theta}' + \boldsymbol{\epsilon}_i)], \text{ with } \boldsymbol{\epsilon}_i \sim \rho(\boldsymbol{\epsilon}) \text{ for } i \in \{1, \ldots, S\}. \tag{9}$$

If the states of both the distributions $p$ and $q$ are binary vectors, $\widehat{\nabla}_{\boldsymbol{\theta}}\mathcal{L}(\boldsymbol{\theta}, \boldsymbol{\theta}') \in [-1, 1]^m$ and when $S = 1$ $\widehat{\nabla}_{\boldsymbol{\theta}}\mathcal{L}(\boldsymbol{\theta}, \boldsymbol{\theta}') \in \{-1, 0, 1\}^m$. In the following, we discuss the problem of constructing families of target distributions $q$. We will also analyze under what assumptions the inequality of Eq. (5) holds.

### 3.1 Target Distributions via Perturbation-based Implicit Differentiation

The efficacy of the I-MLE estimator hinges on a proper choice of $q$, a hyperparameter of our framework. In this section we derive and motivate a class of general-purpose target distributions, rooted in perturbation-based implicit differentiation (PID):

$$q(\boldsymbol{z}; \boldsymbol{\theta}') = p(\boldsymbol{z}; \boldsymbol{\theta} - \lambda\nabla_{\boldsymbol{z}}\ell(f_{\boldsymbol{u}}(\overline{\boldsymbol{z}}), \hat{\boldsymbol{y}})) \text{ with } \overline{\boldsymbol{z}} = \texttt{MAP}(\boldsymbol{\theta} + \boldsymbol{\epsilon}) \text{ and } \boldsymbol{\epsilon} \sim \rho(\boldsymbol{\epsilon}), \tag{10}$$

---

[4]We expand on this in Appendix A where we also review the classic MLE setup [Murphy, 2012, Ch. 9].
[5]This is known as the *perceptron learning rule* in standard MLE.

where $\boldsymbol{\theta} = h_{\boldsymbol{v}}(\hat{\boldsymbol{x}})$, $(\hat{\boldsymbol{x}}, \hat{\boldsymbol{y}}) \in \mathcal{D}$ is a data point, and $\lambda > 0$ is a hyperparameter that controls the perturbation intensity.

To motivate Eq. (10), consider the setting where the inputs to $f$ are the marginals of $p(\boldsymbol{z}; \boldsymbol{\theta})$ (rather than discrete perturb-and-MAP samples as in Eq. (8)), that is, $\boldsymbol{y} = f_{\boldsymbol{u}}(\boldsymbol{\mu}(\boldsymbol{\theta}))$ with $\boldsymbol{\theta} = h_{\boldsymbol{v}}(\hat{\boldsymbol{x}})$, and redefine the training error $L$ of Eq. (2) accordingly. A seminal result by Domke [2010] shows that, in this case, we can obtain $\nabla_{\boldsymbol{\theta}} L$ by perturbation-based differentiation as:

$$\nabla_{\boldsymbol{\theta}} L(\hat{\boldsymbol{x}}, \hat{\boldsymbol{y}}; \boldsymbol{\omega}) = \lim_{\lambda \to 0} \left\{ \frac{1}{\lambda} \left[ \boldsymbol{\mu}(\boldsymbol{\theta}) - \boldsymbol{\mu} \left( \boldsymbol{\theta} - \lambda \nabla_{\boldsymbol{\mu}} L(\hat{\boldsymbol{x}}, \hat{\boldsymbol{y}}; \boldsymbol{\omega}) \right) \right] \right\}, \qquad (11)$$

where $\nabla_{\boldsymbol{\mu}} L = \partial_{\boldsymbol{\mu}} f_{\boldsymbol{u}}(\boldsymbol{\mu})^{\mathsf{T}} \nabla_{\boldsymbol{y}} \ell(\boldsymbol{y}, \hat{\boldsymbol{y}})$. The expression inside the limit may be interpreted as the gradient of an implicit MLE objective (see Eq. (7)) between the distribution $p$ with (current) parameters $\boldsymbol{\theta}$ and $p$ with parameters perturbed in the negative direction of the downstream gradient $\nabla_{\boldsymbol{\mu}} L$. Now, we can adapt (11) to our setting of Eq. (8) by resorting to the straight-through estimator (STE) assumption [Bengio et al., 2013]. Here, the STE assumption translates into reparameterizing $\boldsymbol{z}$ as a function of $\boldsymbol{\mu}$ and approximating $\partial_{\boldsymbol{\mu}} \boldsymbol{z} \approx \boldsymbol{I}$. Then, $\nabla_{\boldsymbol{\mu}} L = \partial_{\boldsymbol{\mu}} \boldsymbol{z}^{\mathsf{T}} \nabla_{\boldsymbol{z}} L \approx \nabla_{\boldsymbol{z}} L$ and we approximate Eq. (11) as:

$$\nabla_{\boldsymbol{\theta}} L(\hat{\boldsymbol{x}}, \hat{\boldsymbol{y}}; \boldsymbol{\omega}) \approx \frac{1}{\lambda} \left[ \boldsymbol{\mu}(\boldsymbol{\theta}) - \boldsymbol{\mu} \left( \boldsymbol{\theta} - \lambda \nabla_{\boldsymbol{z}} L(\hat{\boldsymbol{x}}, \hat{\boldsymbol{y}}; \boldsymbol{\omega}) \right) \right] = \frac{1}{\lambda} \nabla_{\boldsymbol{\theta}} \mathcal{L}(\boldsymbol{\theta}, \boldsymbol{\theta} - \lambda \nabla_{\boldsymbol{z}} L(\hat{\boldsymbol{x}}, \hat{\boldsymbol{y}}; \boldsymbol{\omega})), \quad (12)$$

for some $\lambda > 0$. From Eq. (12) we derive (10) by taking a single sample estimator of $\nabla_{\boldsymbol{z}} L$ (with perturb-and-MAP sampling) and by incorporating the constant $1/\lambda$ into a global learning rate. I-MLE with PID target distributions may be seen as a way to generalize the STE to more complex distributions. Instead of using the gradients $\nabla_{\boldsymbol{z}} L$ to backpropagate directly, I-MLE uses them to construct a target distribution $q$. With that, it defines an implicit maximum likelihood objective, whose gradient (estimator) propagates the supervisory signal upstream, critically, taking the constraints into account. When using Eq. (10) with $\rho(\boldsymbol{\epsilon}) = \delta_0(\boldsymbol{\epsilon})$[6], the I-MLE estimator also recovers a recently proposed gradient estimation rule to differentiate through black-box combinatorial optimization problems [Pogančić et al., 2019]. I-MLE unifies existing gradient estimation rules in one framework. Algorithm 1 shows the pseudo-code of the algorithm implementing Eq. (9) for $S = 1$, using the PID target distribution of Eq. (10). The simplicity of the code also demonstrates that instances of I-MLE can easily be implemented as a layer.

We will resume the discussion about target distributions in Section 4, where we analyze more closely the setup of Example 3. Next, we focus on the perturb-and-MAP strategies and derive a class of noise distributions that is particularly apt to the settings we consider in this work.

## 3.2 A Novel Family of Perturb-and-MAP Noise Distributions

When $p$ is a complex high-dimensional distribution, obtaining Monte Carlo estimates of the gradient in Eq. (7) requires approximate sampling. In this paper, we rely on perturbation-based sampling, also known as perturb and MAP [Papandreou and Yuille, 2011]. In this Section we propose a novel way to design tailored noise perturbations. While the proposed family of noise distributions works with I-MLE, the results of this section are of independent interest and can also be used in other (relaxed) perturb-and-MAP based gradient estimators [e.g. Paulus et al., 2020]. First, we start by revisiting a classic result by Papandreou and Yuille [2011] which theoretically motivates the perturb-and-MAP approach (also known as the Gumbel-max trick), which we generalize here to consider also the temperature parameter $\tau$.

**Proposition 1.** *Let $p(\boldsymbol{z}; \boldsymbol{\theta})$ be a discrete exponential family distribution with integer polytope $\mathcal{C}$ and temperature $\tau$, and let $\langle \boldsymbol{z}, \boldsymbol{\theta} \rangle$ be the unnormalized weight of each $\boldsymbol{z} \in \mathcal{C}$. Moreover, let $\tilde{\boldsymbol{\theta}}$ be such that, for all $\boldsymbol{z} \in \mathcal{C}$, $\langle \boldsymbol{z}, \tilde{\boldsymbol{\theta}} \rangle = \langle \boldsymbol{z}, \boldsymbol{\theta} \rangle + \epsilon(\boldsymbol{z})$ with each $\epsilon(\boldsymbol{z})$ sampled i.i.d. from $\mathrm{Gumbel}(0, \tau)$. Then we have that $\Pr(\mathtt{MAP}(\tilde{\boldsymbol{\theta}}) = \boldsymbol{z}) = p(\boldsymbol{z}; \boldsymbol{\theta})$.*

All proofs can be found in Appendix B. The proposition states that if we can perturb the weights $\langle \boldsymbol{z}, \boldsymbol{\theta} \rangle$ of each $\boldsymbol{z} \in \mathcal{C}$ with independent $\mathrm{Gumbel}(0, \tau)$ noise, then obtaining MAP states from the perturbed model is equivalent to sampling from $p(\boldsymbol{z}; \boldsymbol{\theta})$ at temperature[7] $\tau$. For complex exponential

---

[6]$\delta_0$ is the Dirac delta centered around $0$ – this is equivalent to approximating the marginals with $\mathtt{MAP}$.

[7]Note that the temperature here is different to the temperature of the Gumbel softmax trick [Jang et al., 2017] which scales *both* the sum of the logits *and* the samples from $\mathrm{Gumbel}(0, 1)$.

distributions, perturbing the weights $\langle z, \theta \rangle$ for each state $z \in \mathcal{C}$ is at least as expensive as computing the marginals exactly. Hence, one usually resorts to *local perturbations* of each $[\theta]_i$ (the $i$-th entry of the vector $\theta$) with Gumbel noise. Fortunately, we can prove that, for a large class of distributions, it is possible to design more suitable *local* perturbations. First, we show that, for any $\kappa \in \mathbb{N}^+$, a Gumbel distribution can be written as a finite sum of $\kappa$ i.i.d. (implicitly defined) random variables.

**Lemma 1.** *Let $X \sim \mathrm{Gumbel}(0, \tau)$ and let $\kappa \in \mathbb{N}^+$. Define the Sum-of-Gamma distribution as*

$$\mathrm{SoG}(\kappa, \tau, s) := \frac{\tau}{\kappa} \left\{ \sum_{i=1}^{s} \left\{ \mathrm{Gamma}(1/\kappa, \kappa/i) \right\} - \log(s) \right\}, \tag{13}$$

*where $s \in \mathbb{N}^+$ and $\mathrm{Gamma}(\alpha, \beta)$ is the Gamma distribution with shape $\alpha$ and scale $\beta$, and let $\mathrm{SoG}(\kappa, \tau) := \lim_{s \to \infty} \mathrm{SoG}(\kappa, \tau, s)$. Then we have that $X \sim \sum_{j=1}^{\kappa} \epsilon_j$, with $\epsilon_j \sim \mathrm{SoG}(\kappa, \tau)$.*

Based on Lemma 1, we can show that for exponential family distributions where every $z \in \mathcal{C}$ has exactly $k$ non-zero entries we can design perturbations of $\langle z, \theta \rangle$ following a Gumbel distribution.

**Theorem 1.** *Let $p(z; \theta)$ be a discrete exponential family distribution with integer polytope $\mathcal{C}$ and temperature $\tau$. Assume that if $z \in \mathcal{C}$ then $\langle z, \mathbf{1} \rangle = k$ for some constant $k \in \mathbb{N}^+$. Let $\tilde{\theta}$ be the perturbation obtained by $[\tilde{\theta}]_j = [\theta]_j + \epsilon_j$ with $\epsilon_j \sim \mathrm{SoG}(k, \tau)$ from Eq. (13). Then, $\forall z \in \mathcal{C}$ we have that $\langle z, \tilde{\theta} \rangle = \langle z, \theta \rangle + \epsilon(z)$, with $\epsilon(z) \sim \mathrm{Gumbel}(0, \tau)$.*

Many problems such as $k$-subset selection, traveling salesman, spanning tree, and graph matching strictly satisfy the assumption of Theorem 1. We can, however, also apply the strategy in cases where the variance of $\langle \mathbf{Z}, \mathbf{1} \rangle$ is small (e.g. shortest weighted path). The Sum-of-Gamma perturbations provide a more fine-grained approach to noise perturbations. For $\tau = \kappa = 1$, we obtain the standard Gumbel perturbations. In contrast to the standard $\mathrm{Gumbel}(0, 1)$ noise, the pro-

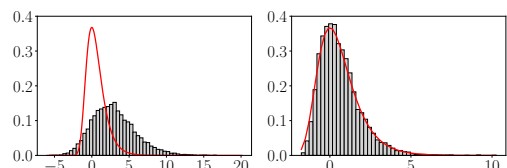

Figure 2: Histograms for 10k samples where each sample is (left) the sum of $5$ $\epsilon_j \sim \mathrm{Gumbel}(0, 1)$ or (right) the sum of $5$ $\epsilon_j \sim \mathrm{SoG}(5, 1, 10)$.

posed local Sum-of-Gamma perturbations result in weights' perturbations that follow the Gumbel distribution. Fig. 2 shows histograms of 10k samples, where each sample is either the sum of 5 samples from $\mathrm{Gumbel}(0, 1)$ (the standard approach) or the sum of $k = 5$ samples from $\mathrm{SoG}(5, 1, 10) = \frac{1}{5} \sum_{i=1}^{10} \{\mathrm{Gamma}(1/5, 5/i) - \log(10)\}$. While we still cannot sample faithfully from $p(z; \theta)$ as the perturbations are *not* independent, we can counteract the problem of partially dependent perturbations by increasing the temperature $\tau$ and, therefore, the variance of the noise distribution. We explore and verify the importance of tuning $\tau$ empirically. In the appendix, we also show that the infinite series from Lemma 1 can be well approximated by a finite sum using convergence results for the Euler-Mascheroni series [Mortici, 2010].

## 4 Target Distributions for Combinatorial Optimization Problems

In this section, we explore the setting where the discrete computational component arises from a combinatorial optimization (CO) problem, specifically an integer linear program (ILP). Many authors have recently considered the setup where the CO component occupies the last layer of the model defined by Eq. (1) (where $f_u$ is the identity) and the supervision is available in terms of examples of either optimal solutions [e.g. Pogančić et al., 2019] or optimal cost coefficients (conditioned on the inputs) [e.g. Elmachtoub and Grigas, 2020]. We have seen in Example 3 that we can naturally associate to each ILP a probability distribution (see Eq. (3)) with $\theta$ given by the negative cost coefficients $c$ of the ILP and $\mathcal{C}$ the integral polytope. Letting $\tau \to 0$ is equivalent to taking the MAP in the forward pass. Furthermore, in Section 3.1 we showed that the I-MLE framework subsumes a recently propose method by Pogančić et al. [2019]. Here, instead, we show that, for a certain choice of the target distribution, I-MLE estimates the gradient of an explicit maximum likelihood learning loss $\mathcal{L}$ where the data distribution is ascribed to either (examples of) optimal solutions or optimal cost coefficients.

Let $q(z; \theta')$ be the distribution $p(z; \theta')$, with parameters

$$[\theta']_i := \begin{cases} [\theta]_i & \text{if } [\nabla_z L]_i = 0 \\ -[\nabla_z L]_i & \text{otherwise.} \end{cases} \tag{14}$$

In the first CO setting, we observe training data $\mathcal{D} = \{(\hat{\boldsymbol{x}}_j, \hat{\boldsymbol{y}}_j)\}_{j=1}^N$ where $\hat{\boldsymbol{y}}_j \in \mathcal{C}$ and the loss $\ell$ measures a distance between a discrete $\hat{\boldsymbol{z}}_j \sim p(\boldsymbol{z}; \boldsymbol{\theta}_j)$ with $\boldsymbol{\theta}_j = h_v(\hat{\boldsymbol{x}})$ and a given optimal solution of the ILP $\hat{\boldsymbol{y}}_j$. An example is the Hamming loss $\ell_H$ [Pogančić et al., 2019] defined as $\ell_H(\boldsymbol{z}, \boldsymbol{y}) = \boldsymbol{z} \circ (\boldsymbol{1} - \boldsymbol{y}) + \boldsymbol{y} \circ (\boldsymbol{1} - \boldsymbol{z})$, where $\circ$ denotes the Hadamard (or entry-wise) product.

**Fact 1.** *If one uses $\ell_H$, then* I-MLE *with the target distribution of Eq. (14) and $\rho(\boldsymbol{\epsilon}) = \delta_0$ is equivalent to the perceptron-rule estimator of the MLE objective between $p(\boldsymbol{z}; h_{\boldsymbol{v}}(\hat{\boldsymbol{x}}_j))$ and $\hat{\boldsymbol{y}}_j$.*

It follows that the method by Pogančić et al. [2019] returns, for a large enough $\lambda$, the maximum-likelihood gradients (scaled by $1/\lambda$) approximated by the perceptron rule. The proofs are given in Appendix B.

In the second CO setting, we observe training data $\mathcal{D} = \{(\hat{\boldsymbol{x}}_j, \hat{\boldsymbol{c}}_j)\}_{j=1}^N$, where $\hat{\boldsymbol{c}}_j$ is the optimal cost conditioned on input $\hat{\boldsymbol{x}}_j$. Here, various authors [e.g. Elmachtoub and Grigas, 2020, Mandi et al., 2020, Mandi and Guns, 2020] use as point-wise loss the regret $\ell_R(\boldsymbol{\theta}, \boldsymbol{c}) = \boldsymbol{c}^\top (\boldsymbol{z}(\boldsymbol{\theta}) - \hat{\boldsymbol{z}}^*(\boldsymbol{c}))$ where $\boldsymbol{z}(\boldsymbol{\theta})$ is a state sampled from $p(\boldsymbol{z}; \boldsymbol{\theta})$ (possibly with temperature $\tau \to 0$, that is, a MAP state) and $\hat{\boldsymbol{z}}^*(\boldsymbol{c}) \in \boldsymbol{z}^*(\boldsymbol{c})$ is an optimal state for $\boldsymbol{c}$.

**Fact 2.** *If one uses $\ell_R$ then* I-MLE *with the target distribution of Eq. (14) is equivalent to the perturb-and-MAP estimator of the MLE objective between $p(\boldsymbol{z}; h_{\boldsymbol{v}}(\hat{\boldsymbol{x}}_j))$ and $p(\boldsymbol{z}; -\hat{\boldsymbol{c}}_j)$.*

This last result also implies that when using the target distribution $q$ from (14) in conjunction with the regret, I-MLE performs maximum-likelihood learning minimizing the KL divergence between the current distribution and the distribution whose parameters are the optimal cost.

Moreover, both facts imply that, when sampling from the MAP states of the distribution $q$ defined by Eq. (14), we have that $\ell(\hat{\boldsymbol{z}}, \hat{\boldsymbol{y}}) = 0$ for $\hat{\boldsymbol{z}} \in \text{MAP}(\boldsymbol{\theta}')$. Therefore, $\ell(\hat{\boldsymbol{z}}, \hat{\boldsymbol{y}}) = 0 \le \mathbb{E}_{\hat{\boldsymbol{z}} \sim p(\boldsymbol{z}; \boldsymbol{\theta})} [\ell((\hat{\boldsymbol{z}}, \hat{\boldsymbol{y}})]$, meaning that the inequality of Eq. (5) is satisfied for $\tau \to 0$.

## 5   Related Work

Several papers address the gradient estimation problem for discrete r.v.s, many resorting to relaxations. Maddison et al. [2017], Jang et al. [2017] propose the Gumbel-softmax distribution to relax categorical r.v.s; Paulus et al. [2020] study extensions to more complex probability distributions. The concrete distribution (the Gumbel-softmax distribution) is only directly applicable to categorical variables. For more complex distributions, one has to come up with tailor-made relaxations or use the straight-through or score function estimators (see for instance Kim et al. [2016], Grover et al. [2019]). In our experiments, we compare with the Gumbel-softmax estimator in Figure 4 (left and right). We show that the $k$-subset VAE trained with I-MLE achieves loss values that are similar to those of the categorical (1-subset) VAE trained with the Gumbel-softmax gradient estimator. Tucker et al. [2017], Grathwohl et al. [2018] develop parameterized control variates (the former was named REBAR) based on continuous relaxations for the score-function estimator. In contrast, we focus explicitly on problems where *only* discrete samples are used during training. Moreover, REBAR is tailored to categorical distributions. I-MLE is intended for models with complex distributions (e.g. those with with many constraints).

Approaches that do not rely on relaxations are specific to certain distributions [Bengio et al., 2013, Franceschi et al., 2019, Liu et al., 2019] or assume knowledge of $\mathcal{C}$ [Kool et al., 2020]. We provide a general-purpose framework that does not require access to the linear constraints and the corresponding integer polytope $\mathcal{C}$. Experiments in the next section show that while I-MLE only requires a MAP solver, it is competitive and sometimes outperforms tailor-made relaxations. SparseMAP [Niculae et al., 2018] is an approach to structured prediction and latent variables, replacing the exponential distribution (specifically, the softmax) with a sparser distribution. Similar to our work, it only presupposes the availability of a MAP oracle. LP-SparseMAP [Niculae and Martins, 2020] is an extension that uses a relaxation of the optimization problem rather than a MAP solver. Sparsity can also be exploited for efficient marginal inference in latent variable models [Correia et al., 2020].

A series of works about differentiating through CO problems [Wilder et al., 2019, Elmachtoub and Grigas, 2020, Ferber et al., 2020, Mandi and Guns, 2020] relax ILPs by adding $L^1$, $L^2$ or log-barrier regularization terms and differentiate through the KKT conditions deriving from the application of the cutting plane or the interior-point methods. These approaches are conceptually linked to techniques for differentiating through smooth programs [Amos and Kolter, 2017, Donti et al., 2017,

Agrawal et al., 2019, Chen et al., 2020, Domke, 2012, Franceschi et al., 2018] that arise not only in modelling but also in hyperparameter optimization and meta-learning. Pogančić et al. [2019], Rolínek et al. [2020], Berthet et al. [2020] propose methods that are not tied to a specific ILP solver. As we saw above, the former two, originally derived from a continuous interpolation argument, may be interpreted as special instantiations of I-MLE. The latter addresses the theory of perturbed optimizers and discusses perturb and MAP in the context of the Fenchel-Young loss. All the CO-related works assume that either optimal costs or solutions are given as training data, while I-MLE may be also applied in the absence of such supervision by making use of implicitly generated target distributions. Other authors focus on devising differentiable relaxations for specific CO problems such as SAT [Evans and Grefenstette, 2018] or MaxSAT [Wang et al., 2019]. Machine learning intersects with CO also in other contexts, e.g. in learning heuristics to improve the performances of CO solvers or differentiable models such as GNNs to "replace" them; see Bengio et al. [2020] and references therein.

Direct Loss Minimization [DLM, McAllester et al., 2010, Song et al., 2016] is also related to our work, but the assumption there is that examples of optimal states $\hat{z}$ are given. Lorberbom et al. [2019] extend the DLM framework to discrete VAEs using coupled perturbations. Their approach is tailored to VAEs and not general-purpose. Under a methodological viewpoint, I-MLE inherits from classical MLE [Wainwright and Jordan, 2008] and perturb-and-MAP [Papandreou and Yuille, 2011]. The theory of perturb-and-MAP was used to derive general-purpose upper bounds for log-partition functions [Hazan and Jaakkola, 2012, Shpakova and Bach, 2016].

## 6 Experiments

The set of experiments can be divided into three parts. First, we analyze and compare the behavior of I-MLE with (i) the score function and (ii) the straight-through estimator using a toy problem. Second, we explore the latent variable setting where both $h_v$ and $f_u$ in Eq. (1) are neural networks and the optimal structure is *not* available during training. Finally, we address the problem of differentiating through black-box combinatorial optimization problems, where we use the target distribution derived in Section 4. More experimental details for available in the appendix.

**Synthetic Experiments.** We conducted a series of experiments with a tractable 5-subset distribution (see Example 2) where $\mathbf{z} \in \{0,1\}^{10}$. We set the loss to $L(\boldsymbol{\theta}) = \mathbb{E}_{\hat{z} \sim p(\mathbf{z};\boldsymbol{\theta})}[\|\hat{z} - \mathbf{b}\|^2]$, where $\mathbf{b}$ is a fixed vector sampled from $\mathcal{N}(0, \mathbf{I})$. In Fig. 3 (Top), we plot optimization curves with means and standard deviations, comparing the proposed estimator with the straight-through (STE) and the score function (SFE) estimators. [8] For STE and I-MLE, we use Perturb-and-MAP (PaM) with Gumbel and $\mathrm{SoG}(1, 5, 10)$ noise, respectively. The SFE uses faithful samples and exact marginals (which is feasible only when $m$ is very small) and converges much more slowly than the other methods, while the STE converges to worse solutions than those found using I-MLE. Fig. 3 (Bottom) shows the benefits of using SoG rather than Gumbel perturbations with I-MLE. While the best configurations for both are comparable, SoG noise achieves in average (over 100 runs) strictly better final values of $L$ for more than $50\%$ of the tested configurations (varying $\lambda$ from Eq. (10) and the learning rate) and exhibit smaller variance (see Fig. 6). Additional details and results in Appendix C.1.

**Learning to Explain.** The BEERADVOCATE dataset [McAuley et al., 2012] consists of free-text reviews and ratings for 4 different aspects of beer: appearance, aroma, palate, and taste. Each sentence in the test set has annotations providing the words that best describe the various aspects. Following the experimental setup of recent work [Paulus et al., 2020], we address the problem introduced by the L2X paper [Chen et al., 2018] of learning a distribution over $k$-subsets of words that best explain a given aspect rating. The complexity of the MAP problem for the $k$-subset distribution is linear in $k$. sThe training set has 80k reviews for the aspect APPEARANCE and 70k reviews for all other aspects. Since the original dataset [McAuley et al., 2012] did not provide separate validation and test sets, we compute 10 different evenly sized validation/test splits of the 10k held out set and compute mean and standard deviation over 10 models, each trained on one split. Subset precision was computed using a subset of 993 annotated reviews. We use pre-trained word embeddings from Lei et al. [2016]. Prior work used non-standard neural networks for which an implementation is not available [Paulus

---

[8]Hyperparameters are optimized against $L$ for all methods independently. Statistics are over 100 runs. We found STE slightly better with Gumbel rather than SoG noise. SFE failed with all tested PaM strategies.

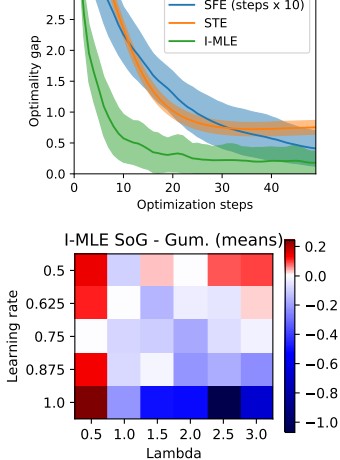

Figure 3: Top: Gradient-based optimization of $L$ with various estimators. Bottom: Mean difference of the final value of $L$ between I-MLE with SoG or Gumbel $\rho(\epsilon)$, varying $\lambda$ and the learning rate (blue = better SoG).

| Method | Test MSE | | Subset Precision | |
|---|---|---|---|---|
| | **Mean** | **Std. Dev.** | **Mean** | **Std. Dev.** |
| $k = 10$ | | | | |
| L2X ($t = 0.1$) | 6.68 | 1.08 | 26.65 | 9.39 |
| SoftSub ($t = 0.5$) | **2.67** | 0.14 | 44.44 | 2.27 |
| STE ($\tau = 30$) | 4.44 | 0.09 | 38.93 | 0.14 |
| I-MLE MAP | 4.08 | 0.91 | 14.55 | 0.04 |
| I-MLE Gumbel | **2.68** | 0.10 | 39.28 | 2.62 |
| I-MLE ($\tau = 30$) | **2.71** | 0.10 | **47.98** | 2.26 |
| $k = 5$ | | | | |
| L2X ($t = 0.1$) | 5.75 | 0.30 | 33.63 | 6.91 |
| SoftSub ($t = 0.5$) | **2.57** | 0.12 | **54.06** | 6.29 |
| I-MLE ($\tau = 5$) | **2.62** | 0.05 | **54.76** | 2.50 |
| $k = 15$ | | | | |
| L2X ($t = 0.1$) | 7.71 | 0.64 | 23.49 | 10.93 |
| SoftSub ($t = 0.5$) | **2.52** | 0.07 | 37.78 | 1.71 |
| I-MLE ($\tau = 30$) | 2.91 | 0.18 | **39.56** | 2.07 |

Table 1: Detailed results for the aspect AROMA. Test MSE and subset precision, both $\times 100$, for $k \in \{5, 10, 15\}$.

et al., 2020]. Instead, we used the neural network from the L2X paper with $4$ convolutional and one dense layer. This neural network outputs the parameters $\boldsymbol{\theta}$ of the distribution $p(\boldsymbol{z}; \boldsymbol{\theta})$ over $k$-hot binary latent masks with $k \in \{5, 10, 15\}$. We compare to relaxation-based baselines L2X [Chen et al., 2018] and SoftSub [Xie and Ermon, 2019]. We also compare the straight-through estimator (STE) with Sum-of-Gamma (SoG) perturbations. We used the standard hyperparameter settings of Chen et al. [2018] and choose the temperature parameter $t \in \{0.1, 0.5, 1.0, 2.0\}$. For I-MLE we choose $\lambda \in \{10^1, 10^2, 10^3\}$, while for both I-MLE and STE we choose $\tau \in \{k, 2k, 3k\}$ based on the validation MSE. We used the standard Adam settings. We trained separate models for each aspect using MSE as point-wise loss $\ell$.

Table 1 lists detailed results for the aspect AROMA. I-MLE's MSE values are competitive with those of the best baseline, and its subset precision is significantly higher than all other methods (for $\tau = 30$). Using only MAP as the approximation of the marginals leads to poor results. This shows that using the tailored perturbations with tuned temperature is crucial to achieve state of the art results. The Sum-of-Gamma perturbation introduced in this paper outperforms the standard local Gumbel perturbations. More details and results can be found in the appendix.

**Discrete Variational Auto-Encoder.** We evaluate various perturbation strategies for a discrete $k$-subset Variational Auto-Encoder (VAE) and compare them to the straight-through estimator (STE) and the Gumbel-softmax trick. The latent variables model a probability distribution over $k$-subsets of (or top-$k$ assignments too) binary vectors of length 20. The special case of $k = 1$ is equivalent to a categorical variable with 20 categories. For $k > 1$, we use I-MLE using the class of PID target distributions of Eq. (10) and compare various perturb-and-MAP noise sampling strategies. The experimental setup is similar to those used in prior work on the Gumbel softmax tricks [Jang et al., 2017]. The loss is the sum of the reconstruction losses (binary cross-entropy loss on output pixels) and the KL divergence between the marginals of the variables and the uniform distribution. The encoding and decoding functions of the VAE consist of three dense layers (encoding: 512-256-20x20; decoding: 256-512-784). We do not use temperature annealing. Using Eq. (9) with $S = 1$, we use either $\text{Gumbel}(0, 1)$ perturbations (the standard approach)[9] or Sum-of-Gamma (SoG) perturbations at a temperature of $\tau = 10$. We run 100 epochs and record the loss on the test data. The difference in training time is negligible. Fig. 4 shows that using the SoG noise distribution is beneficial. The test loss using the SoG perturbations is lower despite the perturbations having higher variance and, therefore, samples of the model being more diverse. This shows that using perturbations of the weights that follow a proper Gumbel distribution is indeed beneficial. I-MLE significantly outperforms the

---

[9]Increasing the temperature $\tau$ of $\text{Gumbel}(0, \tau)$ samples increased the test loss.

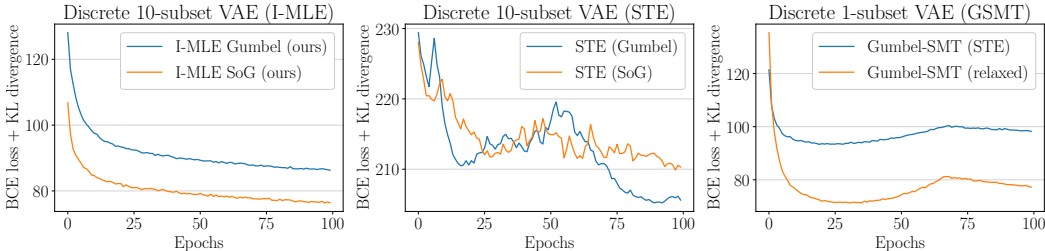

Figure 4: Plots of the sum of the binary reconstruction loss and the KL divergence as a function of the number of epochs (lower is better). (Left) Discrete 10-subset VAE trained with I-MLE with $\lambda = 10$ (I-MLE). (Center) Discrete 10-subset VAE trained with the straight-through estimator (STE). (Right) Discrete 1-subset VAE using the Gumbel softmax trick (GSMT). The down-up-down artifact is due to temperature annealing. Sum-of-Gamma (SoG) perturbations have the lowest test loss for the 10-subset VAEs. For $\lambda = 10$ and SoG perturbations, the test loss is similar to that of the categorical (1-subset) VAE trained with the Gumbel softmax trick.

STE, which does not work in this setting and is competitive with the Gumbel-Softmax trick for the 1-subset (categorical) distribution where marginals can be computed in closed form.

**Differentiating through Combinatorial Solvers.** In these experiments, proposed by Pogančić et al. [2019], the training datasets consists of 10,000 examples of randomly generated images of terrain maps from the Warcraft II tile set [Guyomarch, 2017]. Each example has an underlying $K \times K$ grid whose cells represent terrains

Table 2: Results for the Warcraft shortest path task. Reported is the accuracy, i.e. percentage of paths with the optimal costs. Standard deviations are over five runs.

| $K$ | I-MLE ($\boldsymbol{\mu}$-$\boldsymbol{\mu}$) | I-MLE (M-M) | BB | DPO |
|---|---|---|---|---|
| 12 | $\mathbf{97.2} \pm 0.5$ | $95.2 \pm 0.3$ | $95.2 \pm 0.7$ | $94.8 \pm 0.3$ |
| 18 | $\mathbf{95.8} \pm 0.7$ | $94.4 \pm 0.5$ | $94.7 \pm 0.4$ | $92.3 \pm 0.8$ |
| 24 | $\mathbf{94.3} \pm 1.0$ | $93.2 \pm 0.2$ | $93.8 \pm 0.3$ | $91.5 \pm 0.4$ |
| 30 | $93.6 \pm 0.4$ | $\mathbf{93.7} \pm 0.6$ | $93.6 \pm 0.5$ | $91.5 \pm 0.8$ |

with a fixed cost. The shortest (minimum cost) path between the top-left and bottom-right cell in the grid is encoded as an indicator matrix and serves as the target output. An image of the terrain map is presented to a CNN, which produces a $K \times K$ matrix of vertex costs. These costs are then given to *Dijkstra's algorithm* (the MAP solver) to compute the shortest path. We closely follow the evaluation protocol of Pogančić et al. [2019]. We considered two instantiations of I-MLE: one derived from Fact 1 (M-M in Table 2) using $\ell_H$ and one derived from Fact 2 ($\boldsymbol{\mu}$-$\boldsymbol{\mu}$) using $\ell_R$, with $\rho(\epsilon) = \mathrm{SoG}(k, 1, 10)$ where $k$ is the empirical mean of the path lengths (different for each grid size $K$). We compare with the method proposed by Pogančić et al. [2019] (BB[10]) and Berthet et al. [2020] (DPO). The results are listed in Table 2. I-MLE obtains results comparable to (BB) with M-M and is more accurate with $\boldsymbol{\mu}$-$\boldsymbol{\mu}$. We believe that the $\boldsymbol{\mu}$-$\boldsymbol{\mu}$ advantage may be partially due to an implicit form of data augmentation since we know from Fact 2 that, by using I-MLE, we obtain samples from the distribution whose parameters are the optimal cost. Training dynamics, showing faster convergence of I-MLE ($\boldsymbol{\mu}$-$\boldsymbol{\mu}$), and additional details are available in Table 4.

## 7   Conclusions

I-MLE is an efficient, simple-to-implement, and general-purpose framework for learning hybrid models. I-MLE is competitive with relaxation-based approaches for discrete latent-variable models and with approaches to backpropagate through CO solvers. Moreover, we showed empirically that I-MLE outperforms the straight-through estimator. A limitation of the work is its dependency on computing MAP states which is, in general, an NP-hard problem (although for many interesting cases there are efficient algorithms). Future work includes devising target distributions when $\nabla_{\boldsymbol{z}} L$ is not available, studying the properties (including the bias) of the proposed estimator, developing adaptive strategies for $\tau$ and $\lambda$, and integrating and testing I-MLE in several challenging application domains.

---

[10]Note that this is the same as using I-MLE with PID target distribution form Eq. (10) and $\rho(\epsilon) = \delta_0$.

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
