# A  Standard Maximum Likelihood Estimation and Links to I-MLE

In the standard MLE setting [see, e.g., Murphy, 2012, Ch. 9] we are interested in learning the parameters of a probability distribution, here assumed to be from the (constrained) exponential family (see Eq. (3)), given a set of example states. More specifically, given training data $\mathcal{D} = \{\hat{\mathbf{z}}_j\}_{j=1}^N$, with $\hat{\mathbf{z}}_j \in \mathcal{C} \subseteq \{0,1\}^m$, maximum-likelihood learning aims to minimize the empirical risk

$$\mathcal{L}(\boldsymbol{\theta}, q_{\mathcal{D}}) = \mathbb{E}_{\hat{\mathbf{z}} \sim q_{\mathcal{D}}}[-\log p(\hat{\mathbf{z}}; \boldsymbol{\theta})] = \frac{1}{N} \sum_{j=1}^N -\log p(\hat{\mathbf{z}}_j; \boldsymbol{\theta}) = \frac{1}{N} \sum_{j=1}^N (A(\boldsymbol{\theta}) - \langle \hat{\mathbf{z}}_j, \boldsymbol{\theta} \rangle) \qquad (15)$$

with respect to $\boldsymbol{\theta}$, where $q_{\mathcal{D}}(\boldsymbol{z}) = \sum_j \delta_{\hat{\mathbf{z}}_j}(\boldsymbol{z})/N$ is the empirical data distribution and $\delta_{\hat{\mathbf{z}}}$ is the Dirac delta centered in $\hat{\mathbf{z}}$. In Eq. (15), the point-wise loss $\ell$ is the negative log likelihood $-\log p(\hat{\mathbf{z}}, \boldsymbol{\theta})$, and $q_{\mathcal{D}}$ may be seen as a (data/empirical) *target distribution*. Note that in the main paper, as we assumed $q$ to be from the exponential family with parameters $\boldsymbol{\theta}'$, we used the notation $\mathcal{L}(\boldsymbol{\theta}, \boldsymbol{\theta}')$ to indicate the MLE objective rather than $\mathcal{L}(\boldsymbol{\theta}, q)$. These two definitions are, however, essentially equivalent.

Eq. (15) is a smooth objective that can be optimized with a (stochastic) gradient descent procedure. For a data point $\hat{\mathbf{z}}$, the gradient of the point-wise loss is given by $\nabla_{\boldsymbol{\theta}} \ell = \boldsymbol{\mu}(\boldsymbol{\theta}) - \hat{\mathbf{z}}$, since $\nabla_{\boldsymbol{\theta}} A = \boldsymbol{\mu}$. For the entire dataset one has

$$\nabla_{\boldsymbol{\theta}} \mathcal{L}(\boldsymbol{\theta}, q_{\mathcal{D}}) = \boldsymbol{\mu}(\boldsymbol{\theta}) - \frac{1}{N} \sum_{j=1}^N \hat{\boldsymbol{z}}_j = \boldsymbol{\mu}(\boldsymbol{\theta}) - \mathbb{E}_{\hat{\mathbf{z}} \sim q_{\mathcal{D}}}[\hat{\boldsymbol{z}}_j] \qquad (16)$$

which is (cf. Eq. (7)) the difference between the marginals of $p(\boldsymbol{z}; \boldsymbol{\theta})$ (the mean of $\mathbf{Z}$) and the empirical mean of $\mathcal{D}$, $\boldsymbol{\mu}(q_{\mathcal{D}}) = \mathbb{E}_{\hat{\mathbf{z}} \sim \mathcal{D}}[\hat{\boldsymbol{z}}]$. As mentioned in the main paper, the main computational challenge when evaluating Eq. (16) is to compute the marginals (of $p(\boldsymbol{z}; \boldsymbol{\theta})$). There are many approximate schemes, one of which is the so-called *perceptron rule*, which approximate Eq. (16) as

$$\widehat{\nabla}_{\boldsymbol{\theta}} \mathcal{L}(\boldsymbol{\theta}, q_{\mathcal{D}}) = \texttt{MAP}(\boldsymbol{\theta}) - \frac{1}{N} \sum_{j=1}^N \hat{\boldsymbol{z}}_j$$

and it is frequently employed in a stochastic manner by sampling one or more points form $\mathcal{D}$, rather than computing the full dataset mean.

We may interpret the standard MLE setting described in this section from the perspective of the problem setting we presented in Section 2. The first indeed amounts to the special case of the latter where there are no inputs ($\mathcal{X} = \emptyset$), the (target) output space coincides with the state space of the distribution ($\mathcal{Y} = \mathcal{Z}$), $f$ is the identity mapping, $\ell$ is the negative log-likelihood and the model's parameter coincide with the distribution parameters, that is $\boldsymbol{\omega} = \boldsymbol{\theta}$.

# B  Proofs of Section 3.2 and Section 4

This section contains the proofs of the results relative to the perturb and map section (Section 3.2) and the section on optimal target distributions for typical loss functions when backpropagating through combinatorial optimization problems (section 4). We repeat the statements here, for the convenience of the reader.

**Proposition 1.** *Let $p(\boldsymbol{z}; \boldsymbol{\theta})$ be a discrete exponential family distribution with constraints $\mathcal{C}$ and temperature $\tau$, and let $\langle \boldsymbol{\theta}, \boldsymbol{z} \rangle$ be the unnormalized weight of each $\boldsymbol{z}$ with $\boldsymbol{z} \in \mathcal{C}$. Moreover, let $\tilde{\boldsymbol{\theta}}$ be such that, for all $\boldsymbol{z} \in \mathcal{C}$,*

$$\langle \boldsymbol{z}, \tilde{\boldsymbol{\theta}} \rangle = \langle \boldsymbol{z}, \boldsymbol{\theta} \rangle + \epsilon(\boldsymbol{z})$$

*with each $\epsilon(\boldsymbol{z})$ i.i.d. samples from* $\mathrm{Gumbel}(0, \tau)$. *Then,*

$$\Pr\left(\texttt{MAP}(\tilde{\boldsymbol{\theta}}) = \boldsymbol{z}\right) = p(\boldsymbol{z}; \boldsymbol{\theta}).$$

*Proof.* Let $\epsilon_i \sim \mathrm{Gumbel}(0, \tau)$ i.i.d. and $\tilde{\theta}_i = \theta_i + \epsilon_i$. Following a derivation similar to one made in Papandreou and Yuille [2011], we have:

$$\Pr\{\arg\max(\tilde{\theta}_1, \ldots, \tilde{\theta}_m) = n\} =$$

$$= \Pr\{\tilde{\theta}_n \geq \max_{j \neq n}\{\tilde{\theta}_j\}\}$$

$$= \int_{-\infty}^{+\infty} g(t; \theta_n) \prod_{j \neq n} G(t; \theta_j)\, dt$$

$$= \int_{-\infty}^{+\infty} \frac{1}{\tau} \exp\left(\frac{\theta_n - t}{\tau} - e^{\frac{\theta_n - t}{\tau}}\right) \prod_{j \neq n} \exp\left(-e^{\frac{\theta_j - t}{\tau}}\right) dt$$

$$= \int_{-\infty}^{+\infty} \frac{1}{\tau} e^{\frac{\theta_n - t}{\tau}} \exp\left(-e^{\frac{\theta_n - t}{\tau}}\right) \prod_{j \neq n} \exp\left(-e^{\frac{\theta_j - t}{\tau}}\right) dt$$

$$= \int_0^1 \prod_{j \neq n} z^{\exp\left(\frac{\theta_j - \theta_n}{\tau}\right)} dz \qquad \text{with } z \triangleq \exp\left(-e^{\frac{\theta_n - t}{\tau}}\right)$$

$$= \frac{1}{1 + \sum_{j \neq n} e^{\frac{\theta_j - \theta_n}{\tau}}}$$

$$= \frac{e^{\frac{\theta_n}{\tau}}}{\sum_{j=1}^m e^{\frac{\theta_j}{\tau}}},$$

where $g$ and $G$ are respectively the Gumbel probability density function and the Gumbel cumulative density function. The proposition now follows from arguments made in Papandreou and Yuille [2011] using the maximal equivalent re-parameterization of $p(\boldsymbol{z}; \boldsymbol{\theta})$ where we specify a parameter $\langle \boldsymbol{\theta}, \boldsymbol{z} \rangle$ for each $\boldsymbol{z}$ with $\mathcal{C}(\boldsymbol{z})$ and perturb these parameters. $\qquad\square$

**Lemma 1.** *Let $X \sim \text{Gumbel}(0, \tau)$ and let $\kappa \in \mathbb{N} \setminus \{0\}$. Then we can write*

$$X \sim \sum_{j=1}^{\kappa} \frac{\tau}{\kappa} \left[ \lim_{s \to \infty} \sum_{i=1}^{s} \{\text{Gamma}(1/\kappa, \kappa/i)\} - \log(s) \right],$$

*where $\text{Gamma}(\alpha, \beta)$ is the gamma distribution with shape $\alpha$ and scale $\beta$.*

*Proof.* Let $\kappa \in \mathbb{N} \setminus \{0\}$ and let $X \sim \text{Gumbel}(0, \tau)$. Its moment generating function has the form

$$\mathbb{E}[\exp(tX)] = \Gamma(1 - \tau t). \tag{17}$$

As mentioned in Johnson and Balakrishnan [p. 443, 1998] we know that we can write the Gamma function as

$$\Gamma(1 - \tau t) = e^{\gamma \tau t} \prod_{i=1}^{\infty} \left(1 - \frac{\tau t}{i}\right)^{-1} e^{\frac{-\tau t}{i}} \tag{18}$$

where $\gamma$ is the Euler-Mascheroni constant. We have that

$$\left(1 - \frac{\tau t}{i}\right)^{-1} = \frac{i}{i - \tau t} = \frac{\frac{i}{\tau}}{\frac{i - \tau t}{\tau}} = \frac{\frac{i}{\tau}}{\frac{i}{\tau} - \frac{\tau t}{\tau}} = \frac{\frac{i}{\tau}}{\frac{i}{\tau} - t}.$$

The last term is the moment generating function of an exponential distribution with scale $\frac{\tau}{i}$. We can now take the logarithm on both sides of Eq. (18) and obtain

$$tX = \gamma \tau t + \lim_{s \to \infty} \sum_{i=1}^{s} \left(t \, \text{Exp}(\tau/i) - \frac{\tau t}{i}\right),$$

where $\mathrm{Exp}(\alpha)$ is the exponential distribution with scale $\alpha$. Hence,

$$
\begin{aligned}
X &\sim \lim_{s\to\infty} \sum_{i=1}^{s} \left( \mathrm{Exp}(\tau/i) - \frac{\tau}{i} \right) + \gamma\tau \\
&= \lim_{s\to\infty} \sum_{i=1}^{s} \left( \mathrm{Exp}(\tau/i) - \frac{\tau}{i} \right) + \tau \lim_{s\to\infty} \sum_{i=1}^{s} \frac{1}{i} - \log(s) \\
&= \lim_{s\to\infty} \sum_{i=1}^{s} \left( \mathrm{Exp}(\tau/i) - \frac{\tau}{i} + \frac{\tau}{i} \right) - \tau \log(s) \\
&= \lim_{s\to\infty} \sum_{i=1}^{s} \mathrm{Exp}(\tau/i) - \tau \log(s)
\end{aligned}
$$

Since $\mathrm{Exp}(\alpha) \sim \mathrm{Gamma}(1,\alpha)$, and due to the scaling and summation properties of the Gamma distribution (with shape-scale parameterization), we can write for all $r > 1$:

$$
\mathrm{Exp}(\alpha) \sim \sum_{j=1}^{r} \mathrm{Gamma}(1/r, \alpha r)/r.
$$

Hence, picking $r = \kappa$ from the hypothesis, we have

$$
\begin{aligned}
X &\sim \lim_{s\to\infty} \left\{ \sum_{i=1}^{s} \sum_{j=1}^{\kappa} \mathrm{Gamma}(1/\kappa, \tau\kappa/i)/\kappa \right\} - \tau \log(s) \\
&= \lim_{s\to\infty} \left\{ \sum_{j=1}^{\kappa} \frac{\tau}{k} \sum_{i=1}^{s} \mathrm{Gamma}(1/\kappa, \kappa/i) \right\} - \sum_{j=1}^{\kappa} \frac{\tau}{\kappa} \log(s) \\
&= \lim_{s\to\infty} \sum_{j=1}^{\kappa} \frac{\tau}{\kappa} \left\{ \left[ \sum_{i=1}^{s} \mathrm{Gamma}(1/\kappa, \kappa/i) \right] - \log(s) \right\} \\
&= \sum_{j=1}^{\kappa} \frac{\tau}{\kappa} \left\{ \lim_{s\to\infty} \left[ \sum_{i=1}^{s} \mathrm{Gamma}(1/\kappa, \kappa/i) \right] - \log(s) \right\}
\end{aligned}
$$

This concludes the proof. Parts of the proof are inspired by a post on stackexchange Xi'an [2016]. $\quad\square$

**Theorem 1.** *Let $p(z; \boldsymbol{\theta})$ be a discrete exponential family distribution with constraints $\mathcal{C}$ and temperature $\tau$, and let $k \in \mathbb{N} \setminus \{0\}$. Let us assume that if $\mathcal{C}(z)$ then $\langle z, \mathbf{1} \rangle = k$. Let $\tilde{\boldsymbol{\theta}}$ be the perturbation obtained by $\tilde{\boldsymbol{\theta}}_i = \boldsymbol{\theta}_i + \epsilon_i$ with*

$$
\epsilon_i \sim \frac{\tau}{k} \left[ \lim_{s\to\infty} \sum_{i=1}^{s} \{\mathrm{Gamma}(1/k, k/i)\} - \log(s) \right], \tag{19}
$$

*where $\mathrm{Gamma}(\alpha, \beta)$ is the gamma distribution with shape $\alpha$ and scale $\beta$. Then, for every $z$ we have that $\langle z, \tilde{\boldsymbol{\theta}} \rangle = \langle z, \boldsymbol{\theta} \rangle + \epsilon(z)$ with $\epsilon(z) \sim \mathrm{Gumbel}(0, \tau)$.*

*Proof.* Since we perturb each $\theta_i$ by $\varepsilon_i$ we have, by assumption, that $\langle \boldsymbol{\theta}, \mathbf{1} \rangle = k$, for every $z$ with $\mathcal{C}(z)$, that

$$
\langle z, \tilde{\boldsymbol{\theta}} \rangle = \langle z, \boldsymbol{\theta} \rangle + \sum_{j=1}^{k} \varepsilon_j. \tag{20}
$$

Since by Lemma 1 we know that $\sum_{j=1}^{k} \varepsilon_i \sim \mathrm{Gumbel}(0, \tau)$, the statement of the theorem follows. $\quad\square$

The following theorem shows that the infinite series from Lemma 1 can be well approximated by a finite sum using convergence results for the Euler-Mascheroni series.

**Theorem 2.** *Let $X \sim \text{Gumbel}(0, \tau)$ and $\tilde{X}(m) \sim \sum_{j=1}^{\kappa} \frac{\tau}{\kappa} [\sum_{i=1}^{m} \{\text{Gamma}(1/\kappa, \kappa/i)\} - \log(m)]$.*
*Then*

$$\frac{\tau}{2(m+1)} < \mathbb{E}[\tilde{X}(m)] - \mathbb{E}[X] < \frac{\tau}{2m}.$$

*Proof.* We have that

$$\mathbb{E}[\tilde{X}(m)] =$$

$$= \mathbb{E}\left[\sum_{j=1}^{\kappa} \frac{\tau}{\kappa} \left[\sum_{i=1}^{m} \{\text{Gamma}(1/\kappa, \kappa/i)\} - \log(m)\right]\right]$$

$$= \sum_{i=1}^{m} \mathbb{E}\left[\text{Gamma}(1/\kappa, \tau\kappa/i)\right] - \tau \log(m)$$

$$= \left[\sum_{i=1}^{m} \frac{1}{\kappa} \frac{\tau\kappa}{i} - \tau \log(m)\right]$$

$$= \tau \left[\sum_{i=1}^{m} \frac{1}{i} - \log(m)\right].$$

If $X \sim \text{Gumbel}(0, \tau)$, we know that $\mathbb{E}[X] = \tau\gamma$. Hence,

$$\mathbb{E}[\tilde{X}(m)] - \mathbb{E}[X] = \tau \left[\sum_{i=1}^{m} \frac{1}{i} - \log(m)\right] - \tau\gamma$$

$$= \tau \left[\sum_{i=1}^{m} \frac{1}{i} - \log(m) - \gamma\right].$$

The theorem now follows from convergence results of the Euler-Mascheroni series Mortici [2010].
$\square$

**Fact 1.** *If one uses $\ell_H$, then* I-MLE *with the target distribution of Eq. (14) and $\rho(\boldsymbol{\epsilon}) = \delta_0(\boldsymbol{\epsilon})$ is equivalent to the perceptron-rule estimator of the MLE objective between $p(\boldsymbol{z}; h_{\boldsymbol{v}}(\hat{\boldsymbol{x}}_j))$ and $\hat{\boldsymbol{y}}_j$.*

*Proof.* Rewriting the definition of the Hamming loss gives us

$$\ell_H(\boldsymbol{z}, \boldsymbol{y}) = \frac{1}{m} \sum_{i=1}^{m} (\boldsymbol{z}_i + \boldsymbol{y}_i - 2\boldsymbol{z}_i\boldsymbol{y}_i).$$

Hence, we have that

$$\nabla_{\boldsymbol{z}_i} \ell_H = \frac{1}{m} (1 - 2\boldsymbol{y}_i).$$

Therefore, $\nabla_{\boldsymbol{z}_i} \ell_H = -\frac{1}{m}$ if $\boldsymbol{y}_i = 1$ and $\nabla_{\boldsymbol{z}_i} H = \frac{1}{m}$ if $\boldsymbol{y}_i = 0$. Since, by definition $\boldsymbol{y} \in \mathcal{C}$, we have that

$$\text{MAP}(-\nabla_{\boldsymbol{z}} \ell_H) = \boldsymbol{y}.$$

Now, when using I-MLE with $S = 1$ and $\rho(\boldsymbol{\epsilon}) = \delta_0(\boldsymbol{\epsilon})$ we approximate the gradients as

$$\widehat{\nabla}_{\boldsymbol{\theta}} \mathcal{L}(\boldsymbol{\theta}, \boldsymbol{\theta}') = \text{MAP}(\boldsymbol{\theta}) - \text{MAP}(\boldsymbol{\theta}') = \text{MAP}(\boldsymbol{\theta}) - \text{MAP}(-\nabla_{\boldsymbol{z}} \ell_H) = \text{MAP}(\boldsymbol{\theta}) - \boldsymbol{y}.$$

This concludes the proof. $\square$

**Fact 2.** *If one uses $\ell_R$ then* I-MLE *with the target distribution of Eq. (14) is equivalent to the perturb-and-MAP estimator of the MLE objective between $p(\boldsymbol{z}; h_{\boldsymbol{v}}(\hat{\boldsymbol{x}}_j))$ and $p(\boldsymbol{z}; -\hat{\boldsymbol{c}}_j)$.*

*Proof.* We have that $\nabla_{\boldsymbol{z}_i} \ell_R = \boldsymbol{c}_i$ for all $i$. Now, when using I-MLE with target distribution $\hat{q}(\boldsymbol{z}; \boldsymbol{\theta}')$ of Eq. (14) (and without loss of generality, for $S = 1$) we have that $\hat{q}(\boldsymbol{z}; \boldsymbol{\theta}') = p(\boldsymbol{z}; -\boldsymbol{c})$, and we approximate the gradients as

$$\widehat{\nabla}_{\boldsymbol{\theta}} \mathcal{L}(\boldsymbol{\theta}, \boldsymbol{\theta}') = \text{MAP}(\boldsymbol{\theta} + \boldsymbol{\epsilon}_i) - \text{MAP}(\boldsymbol{\theta}' + \boldsymbol{\epsilon}_i) = \text{MAP}(\boldsymbol{\theta} + \boldsymbol{\epsilon}_i) - \text{MAP}(-\boldsymbol{c} + \boldsymbol{\epsilon}_i), \text{ where } \boldsymbol{\epsilon}_i \sim \rho(\boldsymbol{\epsilon}).$$

Hence, I-MLE approximates the gradients of the maximum likelihood estimation problem between $p(\boldsymbol{z}; h_{\boldsymbol{v}}(\hat{\boldsymbol{x}}_j))$ and $p(\boldsymbol{z}; -\hat{\boldsymbol{c}}_j)$ using perturb-and-MAP. This concludes the proof. $\square$

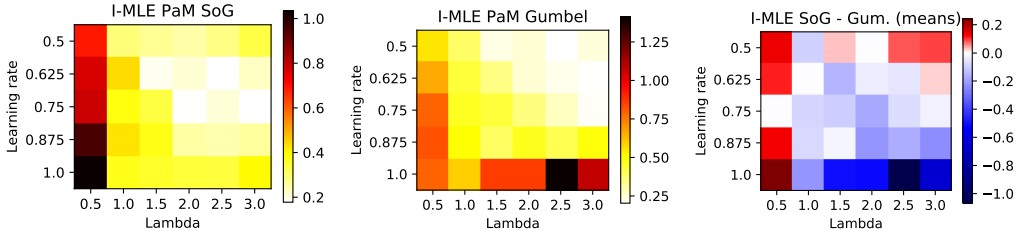

Figure 5: Average (over 100 runs) values of $L(\boldsymbol{\theta})$ after 50 steps of stochastic gradient descent (with momentum) using single-sample I-MLE with $\mathrm{SoG}(1, 5, 10)$ noise (left) and $\mathrm{Gumbel}(0, 1)$ noise (center) varying the perturbation intensity $\lambda$ (see Eq. (10)) and learning rate. The rightmost heat-map depicts the (point-wise) difference between the two methods (blue = better SoG).

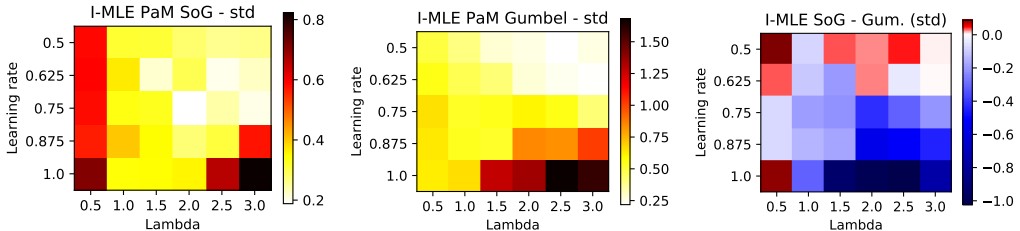

Figure 6: Same as above, but reporting standard deviations.

## C  Experiments: Details and Additional Results

### C.1  Synthetic Experiments

In this series of experiments we analyzed the behaviour of various discrete gradient estimators, comparing our proposed I-MLE with standard straight-trhough (STE) and score-function (SFE) estimators. We also study of the effect of using Sum-of-Gamma perturbations rather than standard Gumbel noise. In order to be able to compute exactly (up to numerical precision) all the quantities involved, we chose a tractable 5-subset distribution (see Example 2) of size $m = 10$.

We set the loss to $L(\boldsymbol{\theta}) = \mathbb{E}_{\hat{\boldsymbol{z}} \sim p(\boldsymbol{z};\boldsymbol{\theta})}[\|\hat{\boldsymbol{z}} - \mathbf{b}\|^2]$, where $\mathbf{b}$ is a fixed vector sampled (only once) from $\mathcal{N}(0, \mathbf{I})$. This amounts to an unconditional setup where there are no input features (as in the standard MLE setting of Appendix A), but where the point-wise loss $\ell(\boldsymbol{z})$ is the Euclidean distance between the distribution output and a fixed vector $\mathbf{b}$. In Fig. 7 we plot the runtime (mean and standard deviation over 10 evaluations) of the full objective $L$ (Expect., in the plot), of a (faithful) sample of $\ell$ and of a perturb-and-MAP sample with Sum-of-Gamma noise distribution (P&M) for increasing size $m$, with $k = m/2$. As it is evident from the plot, the runtime for both expectation and faithful samples, which require computing all the states in $\mathcal{C}$, increases exponentially, while perturb and MAP remains almost constant.

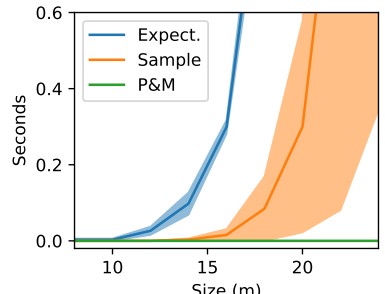

Figure 7: Runtime (mean and standard deviation) for computing $L$ and samples of it, as the dimensionality of the $k$-subset distribution increases (with $k = m/2$).

Within this setting, the one-sample I-MLE estimator is

$$\widehat{\nabla}_{\mathrm{I\text{-}MLE}} L(\boldsymbol{\theta}) = \mathtt{MAP}(\boldsymbol{\theta} + \epsilon) - \mathtt{MAP}(\boldsymbol{\theta}' + \epsilon), \text{ with } \epsilon \sim \mathrm{SoG}(1, 5, 10)$$

where $\boldsymbol{\theta}' = \boldsymbol{\theta} - \lambda[2(\hat{\boldsymbol{z}} - \mathbf{b})]$, where $\hat{\boldsymbol{z}} = \mathtt{MAP}(\boldsymbol{\theta} + \epsilon)$ is a (perturb-and-MAP) sample, while the one-sample straight through estimator is

$$\widehat{\nabla}_{\mathrm{STE}} L(\boldsymbol{\theta}) = 2(\hat{\boldsymbol{z}} - \mathbf{b}), \text{ with } \hat{\boldsymbol{z}} = \mathtt{MAP}(\boldsymbol{\theta} + \epsilon), \ \epsilon \sim \mathrm{Gumbel}(0, 1).$$

For the score function estimator, we have used an expansive faithful sample/full marginal implementation given by

$$\widehat{\nabla}_{\mathrm{SFE}} L(\boldsymbol{\theta}) = \|\hat{\boldsymbol{z}} - \mathbf{b}\|^2 \nabla_{\boldsymbol{\theta}} \log p(\hat{\boldsymbol{z}}; \boldsymbol{\theta}) = \|\hat{\boldsymbol{z}} - \mathbf{b}\|^2 [\hat{\boldsymbol{z}} - \boldsymbol{\mu}(\boldsymbol{\theta})], \text{ with } \hat{\boldsymbol{z}} \sim p(\boldsymbol{z}; \boldsymbol{\theta})$$

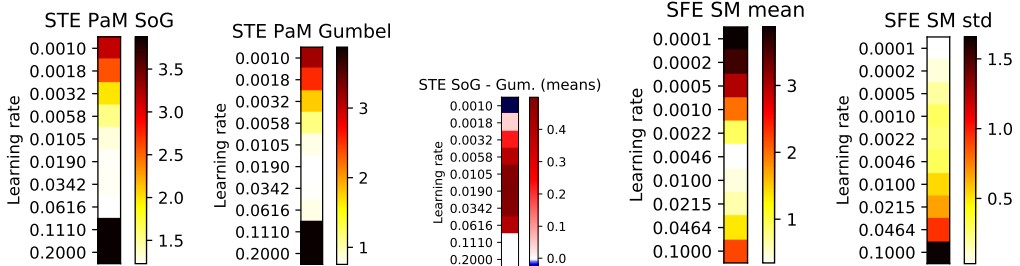

Figure 8: First three plots, from left to right: average (over 100 runs) final values of $L(\boldsymbol{\theta})$ after 50 steps of optimization using the straight-through estimator with SoG noise, varying the learning rate; same but using Gumbel noise; difference of averages between the first and the second heat-maps. Last two plots, from left to right: average (over 20 runs) final values of $L(\boldsymbol{\theta})$ after 500 steps of optimization using the score function estimator (with faithful samples and exact marginals), varying the learning rate; standard deviation for the same setting.

| Method | Appearance | | Palate | | Taste | |
|---|---|---|---|---|---|---|
| | Test MSE | Subset precision | Test MSE | Subset precision | Test MSE | Subset precision |
| L2X ($t = 0.1$) | $10.70 \pm 4.82$ | $30.02 \pm 15.82$ | $6.70 \pm 0.63$ | $50.39 \pm 13.58$ | $6.92 \pm 1.61$ | $32.23 \pm 4.92$ |
| SoftSub ($t = 0.5$) | $\mathbf{2.48} \pm 0.10$ | $52.86 \pm\ \ 7.08$ | $\mathbf{2.94} \pm 0.08$ | $39.17 \pm\ \ 3.17$ | $\mathbf{2.18} \pm 0.10$ | $\mathbf{41.98} \pm 1.42$ |
| I-MLE ($\tau = 30$) | $\mathbf{2.51} \pm 0.05$ | $\mathbf{65.47} \pm\ \ 4.95$ | $\mathbf{2.96} \pm 0.04$ | $40.73 \pm\ \ 3.15$ | $2.38 \pm 0.04$ | $\mathbf{41.38} \pm 1.55$ |

Table 3: Experimental results (mean $\pm$ std. dev.) for the learning to explain experiments for $k = 10$ and various aspects.

since, in preliminary experiments, we did not manage to obtain meaningful results with SFE using perturb-and-MAP for sampling and/or marginals approximation. These equations give the formulae for the estimators which we used for the results plotted in Fig. 3 (top) in the main paper.

In Fig. 5 we plot the heat-maps for the sensitivity results comparing between I-MLE with SoG and I-MLE with Gumbel perturbations. The two leftmost heat-maps depict the average value (over 100 runs) of $L(\boldsymbol{\theta})$ after 50 steps of stochastic gradient descent, for various choices of $\lambda$ and learning rates (momentum factor was fixed at 0.9 for all experiments). The rightmost plot of Fig. 5 is the same as the one in the main paper, and represents the difference between the first and the second heat-maps. Fig. 6 refers to the same setting, but this time showing standard deviations. The rightmost plot of Fig. 6 suggests that using SoG perturbations results also in reduced variance (of the final loss) for most of the tried hyperparameter combinations. Finally, in Fig. 8 we show sensitivity plots for STE (both with SoG and Gumbel perturbations) and SFE, where we vary the learning rate.

## C.2 Learning to Explain

Experiments were run on a server with Intel(R) Xeon(R) CPU E5-2637 v4 @ 3.50GHz, 4 GeForce GTX 1080 Ti, and 128 GB RAM.

The pre-trained word embeddings and data set can be found here: `http://people.csail.mit.edu/taolei/beer/`. Figure 10 depicts the neural network architecture used for the experiments. As in prior work, we use a batch size of 40. The maximum review length is 350 tokens. We use the standard neural network architecture from prior work Chen et al. [2018], Paulus et al. [2020]. The dimensions of the token embeddings (of the embedding layers) are 200. All 1D convolutional layers have 250 filters with a kernel size of 3. All dense layers have a dimension of 100. The dropout layer has a dropout rate of 0.2. The layer Multiply perform the multiplication between the token mask (output of I-MLE) and the embedding matrix. The Lambda layer computes the mean of the selected embedding vectors. The last dense layer has a sigmoid activation. IMLESubsetkLayer is the layer implementing I-MLE. We train for 20 epochs using the standard Adam settings in Tensorflow 2.4.1 (learning rate=0.001, beta1=0.9, beta2=0.999, epsilon=1e-07, amsgrad=False), and no learning rate schedule. The training time (for the 20 epochs) for I-MLE, with sum-of-Gamma perturbations, is 380 seconds, for SoftSub 360 seconds, and for L2X 340 seconds. We always evaluate the model with the best validation MSE among the 20 epochs.

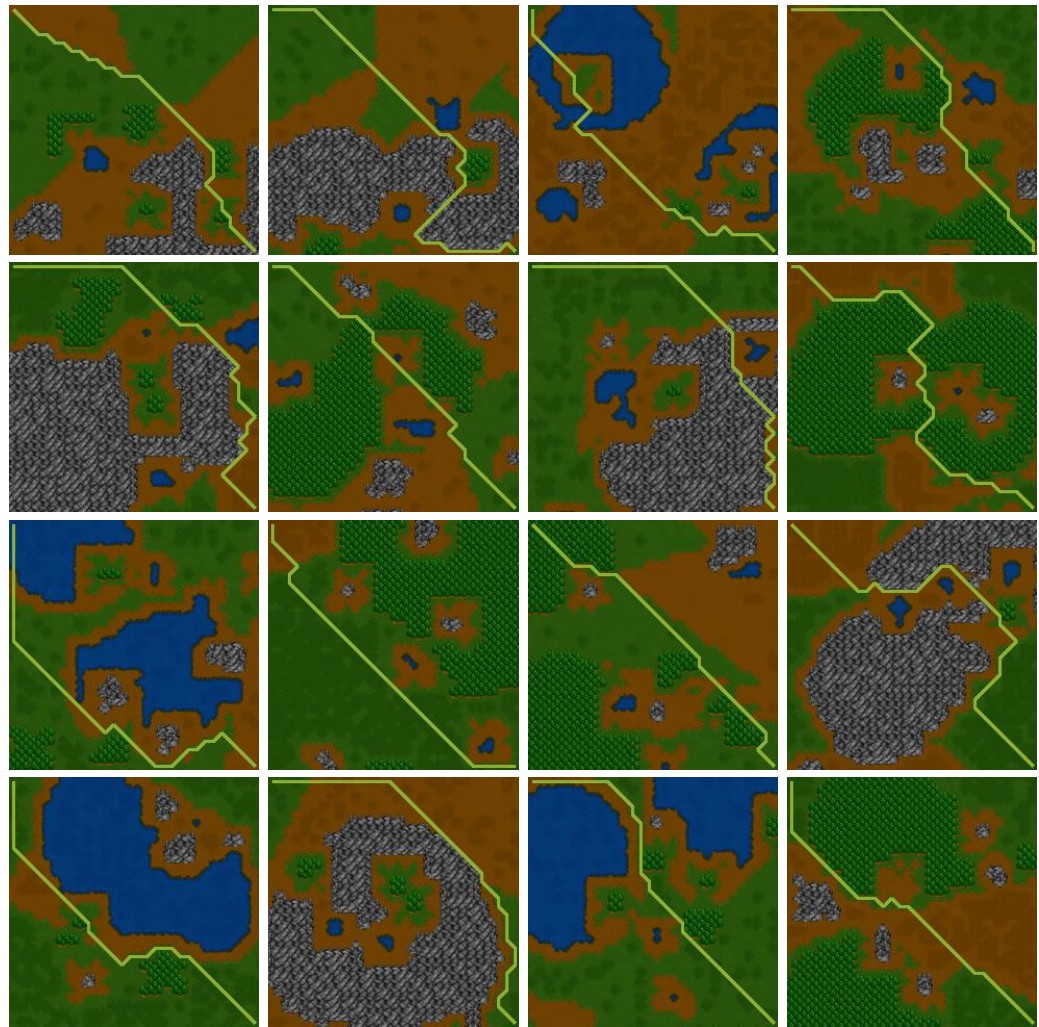

Table 4: Sample of Warcraft maps, and corresponding shortest paths from the upper left to the lower right corner of the map.

Implementations of I-MLE and all experiments will soon be made available. Table 3 lists the results for L2X, SoftSub, and I-MLE for three additional aromas and $k = 10$.

### C.3 Discrete Variational Auto-Encoder

Experiments were run on a server with Intel(R) Xeon(R) CPU E5-2637 v4 @ 3.50GHz, 4 GeForce GTX 1080 Ti, and 128 GB RAM.

The data set can be loaded in Tensorflow 2.x with *tf.keras.datasets.mnist.load_data()*. As in prior work, we use a batch size of 100 and train for 100 epochs, plotting the test loss after each epoch. We use the standard Adam settings in Tensorflow 2.4.1 (learning rate=0.001, beta1=0.9, beta2=0.999, epsilon=1e-07, amsgrad=False), and no learning rate schedule. The MNIST dataset consists in black-and-white $28 \times 28$ pixels images of hand-written digits. The encoder network consists of an input layer with dimension 784 (we flatten the images), a dense layer with dimension 512 and ReLu activation, a dense layer with dimension 256 and ReLu activation, and a dense layer with dimension 400 ($20 \times 20$) which outputs the $\theta$ and no non-linearity. The IMLESubsetkLayer takes $\theta$ as input and outputs a discrete latent code of size $20 \times 20$. The decoder network, which takes this discrete latent code as input, consists of a dense layer with dimension 256 and ReLu activation, a dense layer with dimension 512 and ReLu activation, and finally a dense layer with dimension 784 returning the logits for the output pixels. Sigmoids are applied to these logits and the binary cross-entropy

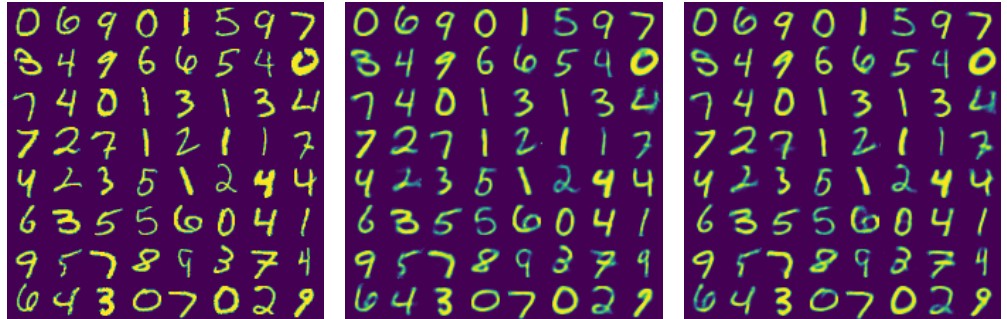

Figure 9: Original MNIST digits from the test set and their reconstructions using the discrete 10-subset VAE trained with Sum-of-Gamma perturbations for $\lambda = 1$ (center) and $\lambda = 10$ (right).

Table 5: Results for the Warcraft shortest path task using I-MLE with two target distributions, namely Eq. (10) and Eq. (14). Reported is the accuracy, i.e. percentage of paths with the optimal costs. Standard deviations are over five runs.

| $K$ | $\mu$-$\mu$, Eq. (14) | M-M, Eq. (14) | $\lambda = 20$, Eq. (10) | $\lambda = 20, \tau = 0.01$, Eq. (10) |
|---|---|---|---|---|
| 12 | $\mathbf{97.2} \pm 0.5$ | $95.2 \pm 0.3$ | $95.2 \pm 0.7$ | $95.1 \pm 0.4$ |
| 18 | $\mathbf{95.8} \pm 0.7$ | $94.4 \pm 0.5$ | $94.7 \pm 0.4$ | $94.4 \pm 0.4$ |
| 24 | $\mathbf{94.3} \pm 1.0$ | $93.2 \pm 0.2$ | $93.8 \pm 0.3$ | $93.7 \pm 0.4$ |
| 30 | $93.6 \pm 0.4$ | $93.7 \pm 0.6$ | $93.6 \pm 0.5$ | $\mathbf{93.8} \pm 0.3$ |

loss is computed. The training time (for the 100 epochs) was 21 minutes with the sum-of-Gamma perturbations and 18 minutes for the standard Gumbel perturbations.

### C.4 Differentiating through Combinatorial Solvers

The experiments were run on a server with Intel(R) Xeon(R) Silver 4208 CPU @ 2.10GHz CPUs, 4 NVIDIA Titan RTX GPUs, and 256 GB main memory.

Table 4 shows a set of $30 \times 30$ Warcraft maps, and the corresponding shortest paths from the upper left to the lower right corner of the map. In these experiments, we follow the same experimental protocol of Pogančić et al. [2019]: optimisation was carried out via the Adam optimiser, with scheduled learning rate drops dividing the learning rate by 10 at epochs 30 and 40. The initial learning rate was $5 \times 10^{-4}$, and the models were trained for 50 epochs using 70 as the batch size. As in [Pogančić et al., 2019], the $K \times K$ weights matrix is produced by a subset of ResNet18 [He et al., 2016], whose weights are trained on the task. For training BB, in all experimental results in Section 6 and Table 4, the hyperparameter $\lambda$ was set to $\lambda = 20$.

Fig. 11 shows the training dynamics of different models, including the method proposed by Pogančić et al. [2019] (BB) with different choices of the $\lambda$ hyperparameter, the ResNet18 baseline proposed by Pogančić et al. [2019], and I-MLE.

Furthermore, we experimented with two different target distributions, namely Eq. (10) and Eq. (14), where noise samples were drawn from a sum-of-Gamma distribution. Results are summarised in Table 5. In our experiments, the two target distributions yield very similar results for $\tau = 0.01$, and results tend to degrade for larger values of $\tau$. This is to be expected, since the target distribution in Eq. (14) is meaningful in the context described in Section 4, where there are forms of explicit supervision over the discrete states. Code and data for all the experiments described in this paper are available online, at `https://github.com/nec-research/tf-imle`.

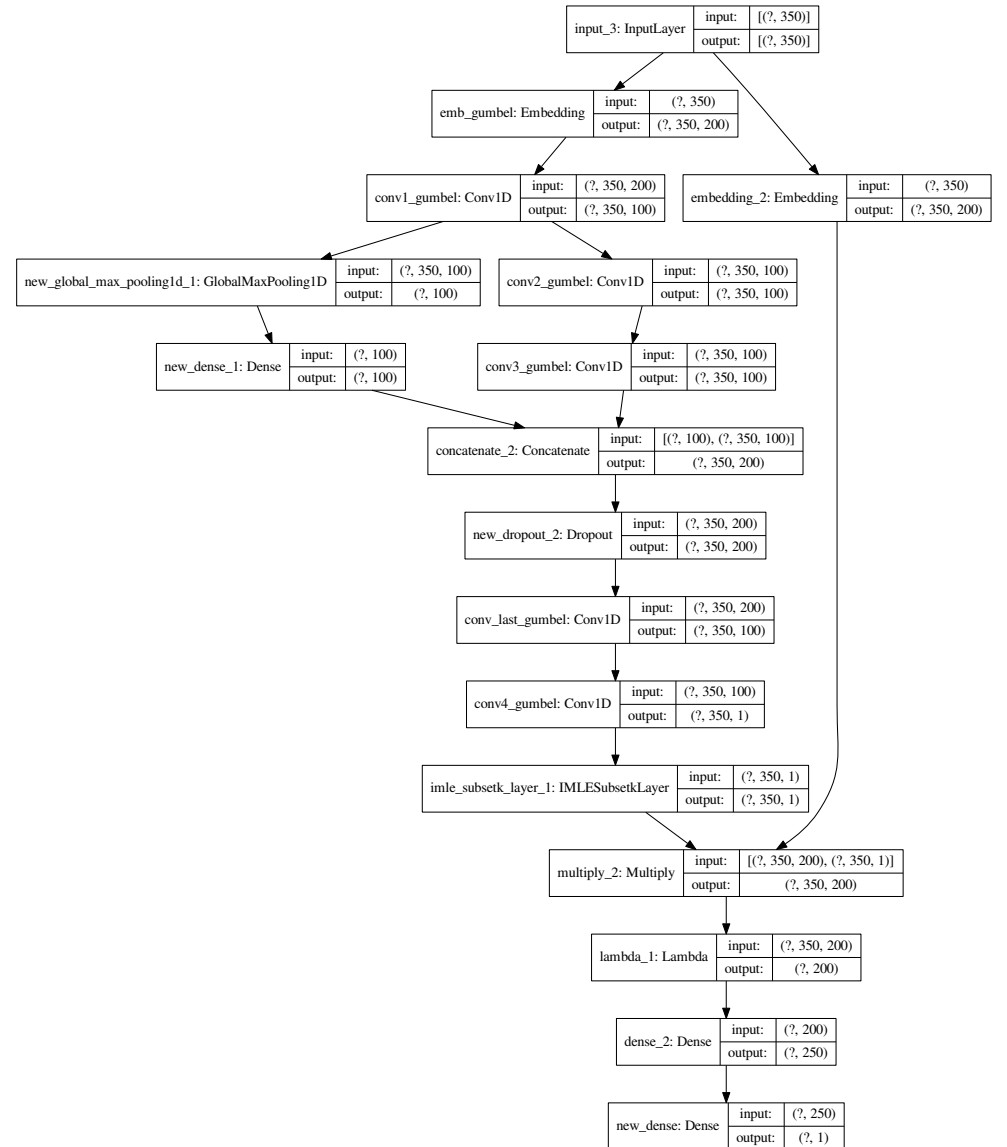

Figure 10: The neural network architecture for the learning to explain experiments. (Please zoom into the vector graphic for more details.) We use the standard architecture and settings from prior work Chen et al. [2018]. The maximum review length is 350 tokens. The dimensions of the token embeddings (of the embedding layers) are 200. All 1D convolutional layers have 250 filters with a kernel size of 3. All dense layers have a dimension of 100. The dropout layer has a dropout rate of 0.2. The layer Multiply perform the multiplication between the token mask (output of I-MLE) and the embedding matrix. The Lambda layer computes the mean of the selected embedding vectors The last dense layer has a sigmoid activation. IMLESubsetkLayer is the layer implementing I-MLE. Code is available in the submission system.

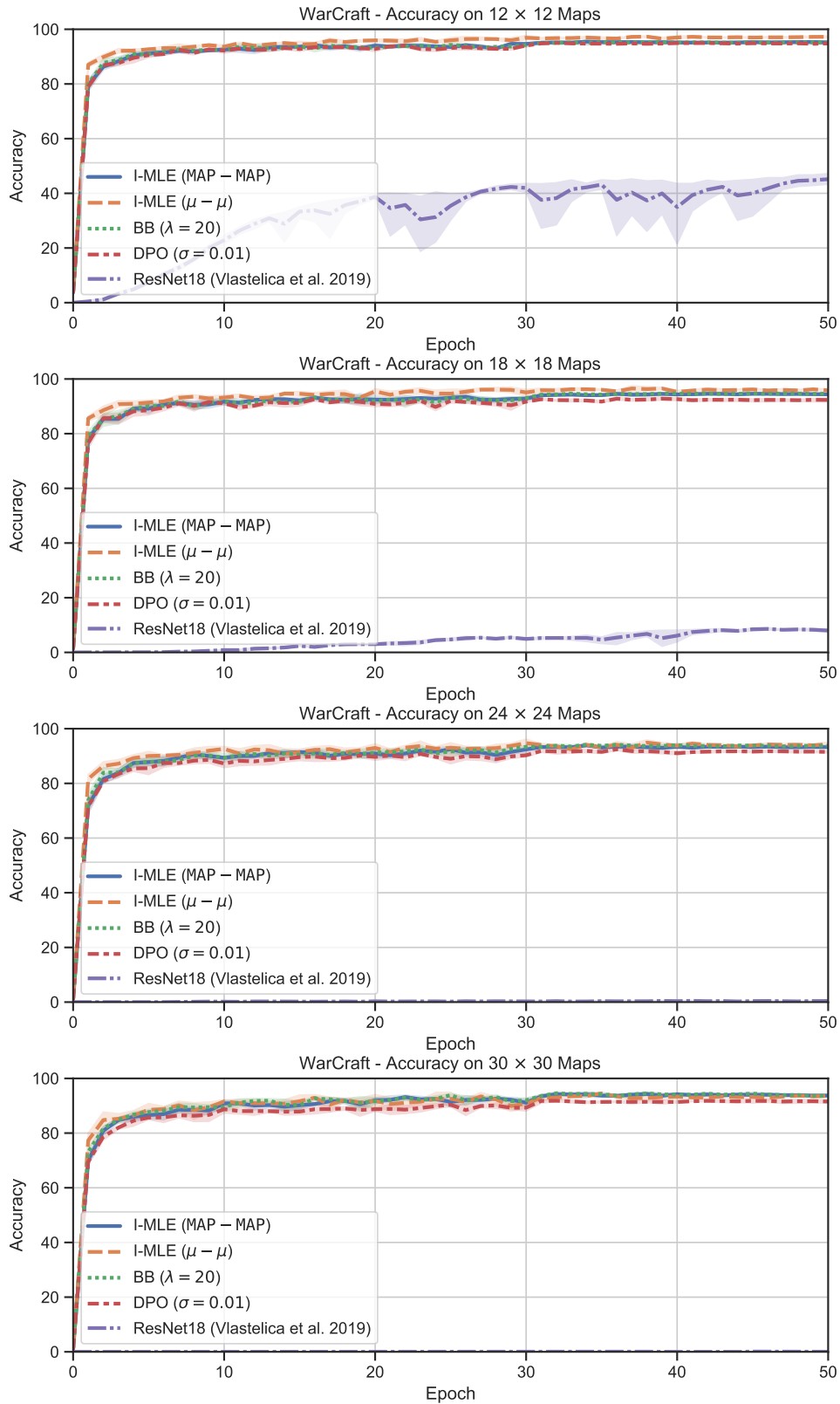

Figure 11: Training dynamics for different models on $K \times K$ shortest path tasks on Warcraft maps, with $K \in \{12, 18, 24, 30\}$.