# OpenReview forum: "Implicit MLE: Backpropagating Through Discrete Exponential Family Distributions"
_NeurIPS.cc/2021/Conference — NeurIPS 2021 Poster_

### Official Review · Reviewer_e8wB · 2021-07-13

**Rating:** 6
**Confidence:** 3

**Summary:**

The authors propose a method for differentiating models that combine neural network components with a discrete distribution. The method estimates gradients by computing most probable states akin to perturb-and-MAP. Simulations on three problems show comparable performance or slight improvement over existing methods.

**Limitations And Societal Impact:**

Yes.

**Main Review:**

The problem of differentiating through discrete representations (e.g., representations of discrete probability distributions) is certainly very relevant.

The paper borrows some ideas from methods such as perturb-and-MAP, and applies them to the setting described in eq. (2) (input feature map -- discrete distribution -- output feature map). The authors contrast their method to be potentially more widely applicable than, say, methods based on smooth relaxations. While this may be true for some classes of problems, I find the requirement to be able to perform MAP inference to be quite a big constraint. Efficient MAP inference is only possible for quite a limited number of discrete distribution families.

I think the biggest problem with this paper is its exposition. This may be due to the fact that I am not an expert is some of the topics covered, but I find that section 3 could have been presented in a much clearer way. As it is, it is quite hard to judge the specific contributions of the paper. The experimental section is better, but even there some parts could have been motivated in a better way (e.g., the VAE application).

Further comments:
- How is this setup related to energy-based models (e.g., see [1])? While I see some basic differences, the form of distributions, and some of the problems considered (e.g., computing gradients of discrete representations) may be very similar.
- Table 1 only reports only results on the "aroma" aspect? How do the other aspects compare?

&nbsp;
### References
[1] Song and Kingma. How to Train Your Energy-Based Models. ArXiv, 2021.

**Time Spent Reviewing:**

5

---

> ### Author Response · Authors · 2021-08-10
> **Answer to Reviewer e8wB**
>
> ​​Thank you for your insightful questions and comments.
>
> > As it is, it is quite hard to judge the specific contributions of the paper.
>
> Thank you for the comment. In the updated version, we will add a paragraph to the introduction where we outline the contributions as follows.  First, we propose I-MLE as a general framework to perform gradient estimation for discrete probability distributions that do not require ad-hoc continuous relaxations. The framework requires two main ingredients: a suitable target distribution, and noise distribution. In Sections 3.1 and 4, we propose two broadly applicable classes of target distributions. In Section 3.2, we propose a novel class of noise distributions tailored to the problem setting. Finally, we empirically show that I-MLE with the proposed target and noise distributions is competitive with and sometimes outperforms existing gradient estimation strategies.
>
> > Efficient MAP inference is only possible for quite a limited number of discrete distribution families.
>
> Indeed, the proposed method relies on the ability of computing MAP inference with a given distribution. However, there are several cases (e.g. when using black-box combinatorial solvers, such as a shortest path algorithm) where computing marginals and sampling is not tractable, while MAP states can be computed efficiently.
> Also, in the Appendix, Section D, we empirically compare the complexity of perturb-and-MAP with a Sum-of-Gamma noise distribution, with faithful sampling and expectations, showing that the proposed method has significantly better sampling properties.
>
> > Exposition: Section 3 could have been presented in a much clearer way.
>
> We strive to make the paper as accessible as possible and are open to further suggestions on how to improve the presentation of Section 3. The rationale for the current structure is to first present the general ideas and framework at the beginning of Section 3 and then focus on the two main sub-problems that emerge from our formulations, namely how to construct suitable target distributions and how to perform perturb-and-MAP sampling. Hence, in Section 3.1 we present a family of general-purpose target distributions rooted in perturbation-based implicit differentiation, and in Section 3.2 we derive a family of noise distributions that are particularly suited to the discrete settings considered in the paper, where often one can know precisely or at least estimate the number of 1s in the discrete states $z$.
>
> > The experimental section is better, but some parts could have been motivated in a better way (e.g., the VAE application).
>
> We will add a more detailed motivation for the VAE application. Here, the idea is that discrete VAEs typically only use categorical distributions. In several applications, however, one might want to use complex constraints and this is made possible by I-MLE. For instance, in a discrete world model [2], one might want to restrict the possible discrete states of an environment using more complex constraints.
>
> [2] Mastering Atari with Discrete World Model's, Danijar Hafner, Timothy Lillicrap, Mohammad Norouzi, Jimmy Ba, ICLR 2021
>
>
> > How is this setup related to energy-based models [...]? While I see some basic differences, the form of distributions and some of the problems considered (e.g., computing gradients of discrete representations) may be very similar.
>
> The main difference is that typically when working with EBMs one “ignores” the partition function and aims to minimize an unnormalized energy function. The methods we presented here, however, could also be used to train normalized EBMs in settings where a proper (normalized) probability distribution is required. We think this is outside the scope of the paper, but consider it as interesting future work. Thanks for the comment.
>
> > Table 1 only reports results on the "aroma" aspect. How do the other aspects compare?
>
> Please note that the results for the other aspects are listed in Table 3, Section D in the appendix.

---

### Official Review · Reviewer_pyap · 2021-07-13

**Rating:** 8
**Confidence:** 3

**Summary:**

The paper proposes a method to compute gradients while training stochastic models with discrete latent variables. The proposed method assumes that latents are drawn from a discrete exponential family distribution: it defines a target distribution over the discrete latents, and uses the target distribution to compute a likelihood which can then be minimised to learn the parameters of the model.

**Limitations And Societal Impact:**



**Main Review:**

The novel approach to estimating gradients for models with discrete latents, as well as the evidence that it outperforms score-function estimators appears to be compelling. Overall the paper was clear and well-written.

However, sections 3 and 4 were quite hard to follow since the discussion jumps back and forth between the gradient estimator / loss function, and the perturb-and-map distributions. For example, $z=MAP(\theta + \epsilon)$ is already defined and used extensively in the introduction to section 3 and in section 3.1, before perturb-and-MAP distributions are discussed in detail and section 4. These sub-sections could be re-arranged and re-written for clarity.

Moreover, in section 3 the loss $l(f_u(z), y)$ is replaced with $L(\theta, \theta')$. The text states that it is intuitive that minimising the latter results in minimising the former.  The text also suggests that the construction of $q(z, \theta')$, the target distribution is a hyperparameter. However, from section 3.1, it seems that the replacement of the loss is possible only when the target distribution $q$ is specifically constructed such that the $l(f_u(z), y)$ is minimised, and that this in turn, hinges on sampling the latent variables using the perturb-and-MAP technique. From this it seems that the "hyperparameter" really is the choice of the noise distribution for the perturbation step, and that the construction of the loss / target distribution is otherwise non-trivial. If this is indeed the case, this should be made clear in the text.

The conclusion claims that I-MLE outperforms relaxation-based techiniques -- this is not obvious from the results.  Although the Gumbel distribution is used in conjunction with the MLE estimator, this does not appear to be the same as the approach in Tucker et al (REBAR) or using concrete relaxed distributions as in Maddison et al. It would be useful to have comparisons between I-MLE and these methods as well.

Minor comments:
1) Figure 1 shows $f_u$ mapping $z$ to $L$ rather than $y$
1) Figure 3 top plot shows optimisation gap versus optimisation steps. Is the optimisation gap the same as the loss function? If not, this performance measure should be explained.
2) Example 3 in section 2 reads more like a theorem -- it would be good to have a clear explanation.


**Time Spent Reviewing:**

5

---

> ### Author Response · Authors · 2021-08-10
> **Answer to Reviewer pyap**
>
> Thank you for your insightful questions and comments.
>
> > Overall the paper was clear and well-written. However, sections 3 and 4 were quite hard to follow [...]
>
> Thank you for your comment -- we will improve the structure of the final version of the paper. We strive to make the paper as accessible as possible and are open to further suggestions.
>
> > Moreover, in section 3 the loss $\ell(fu(z),y)$ is replaced with $\mathcal{L}(\theta, \theta′)$.
>
> Please note that these losses are two different functions: $\ell$ is used to define the training error in Eq. (2), while $\mathcal{L}$ (Section 3) defines the (implicit) MLE objective between the model distribution $p$ and the target distribution $q$.
>
> > The hyperparameter really is the choice of the noise distribution for the perturbation step.
>
> Thank you for pointing this out: the noise distribution is indeed a hyperparameter and we have seen consistently good results for the proposed sum-of-Gamma noise distribution. If we choose the target distribution from 3.1, hyperparameters are lambda and the noise distribution. We will clarify this in the text.
>
> > It would be useful to have comparisons between I-MLE, REBAR (Tucker et al.), and the concrete distribution (Maddison et al.)
>
> The concrete distribution (the Gumbel-softmax trick) is only directly applicable to categorical variables. For more complex distributions, one has to come up with tailor-made relaxations or use the straight-through or score function estimators. In our experiments, we compare with the Gumbel-softmax estimator in Figure 4 (left and right). We show that the k-subset VAE trained with I-MLE achieves loss values that are similar to those of the categorical (1-subset) VAE trained with the Gumbel-softmax gradient estimator. REBAR is also tailored to categorical distributions.
>
> As a general comment, we believe I-MLE is best used when one deals with complex distributions (e.g. with many constraints). If the discrete distribution is categorical (when the marginals can be computed efficiently) then the first choice should be the concrete distribution (Gumbel-softmax trick) and/or REBAR. We will add this comment to the conclusion.
>
> > Figure 1 shows fu mapping z to L rather than y.
>
> Thank you. We will add $y$ before the loss to clarify Figure 1.
>
> > Figure 3 top plot shows optimisation gap versus optimisation steps. Is the optimisation gap the same as the loss function? If not, this performance measure should be explained.
>
> The optimality gap is defined as the difference between the current value of the loss and the lowest achievable loss (the optimal value): that is $L(\theta_t) - L(\theta^*)$. We will make this clear in the caption. The optimality gap is essentially a translated version of the loss which has 0 as a minimum, and it makes it easier to visualize how different methods perform in absolute terms, besides comparing them to each other.
>
> Thank you again for your helpful comments and questions.

---

### Official Review · Reviewer_JsU3 · 2021-07-18

**Rating:** 6
**Confidence:** 3

**Summary:**

The paper proposes a meta algorithm for models involving a discrete bottleneck by making use of a specific target distribution with respect to which a max-likelihood loss is optimized in order to avoid having to compute the high variance score function gradients of the original objective.  This target distribution is characterized by being a local improvement to the overall objective.  To avoid marginalization on the discrete variables, the paper further consider approximations to the proposed I-MLE framework using the perturb-and-MAP approach, which repeatedly solves MAP subroutines for noise-perturbed parameters to approximate the marginals. In doing this, it also proposes a decomposition of Gumbel noise terms into a sum of Gamma distributions.

The ideas are evaluated empirically both on synthetic problems as well as benchmarks involving discrete VAEs and other combinatorial problems requiring flexible discrete latent variable modeling.


**Limitations And Societal Impact:**

The paper is not directly relevant for societal impact issues due to its theoretical nature.

**Main Review:**

This paper considers an approximation to maximum likelihood learning in models that involve a discrete latent variable ($z$) bottleneck between the input features ($x$) and the output ($y$), where $z$ is modeled as being from the discrete exponential family where a parametrized function approximator predicts the natural family parameters for the exponential family.  The last stage of the model maps the realizations of this discrete random variable to the model output. It is assumed that we are given a loss function that depends only on $z, y$, with an overall loss for a given $x, y$ pair being the expectation over the $z$ with a distribution defined by the natural parameters specified by $x$.

Training the model involves learning $\omega = (u, v)$, where the first stage parameters are $v$ and the final stage params are $u$. While a standard score function estimator approach may be enough for learning $u$ (since the partials wrt $u$ are an expectation of gradients, whose distribution does not depend on $u$), the variance for this approach is too high to obtain gradients wrt $v$.  To address this, a target distribution decoupled from $\theta$ is considered, which is only required to strictly improve on the original objective and then the goal is to optimize a maximum likelihood gradient wrt this target distribution. The resulting objective is called the implicit-MLE loss, but clearly a key aspect of this involves defining the target distribution to learn from.

Questions/comments for the authors to clarify:

- A key property that characterizes these target distributions is given by the Equation between lines 116 and 117 (I would strongly encourage the authors to give this Equation a reference as this seems very critical to me).
$$E_{\hat{z} \sim q(z; \theta')} l(f_u(\hat{z}), \hat{y})  \le E_{\hat{z} \sim p(z; \theta)} l(f_u(\hat{z}), \hat{y})$$
But the above equation does not seem to be actually verified in any of the proposals constructed by the authors both in Sections 3.1 or in Section 4. Could the authors please elaborate?

- In Eq (8), the target $q(z, \theta')$ definition seems ambiguous as the parameters on the RHS depend on $z$, which is the variable whose distribution is being specified. Could you please clarify the precise definition for $q$ in this Equation?

- In Eq (12), it appears that target distribution $q$ has the same form as $p$, except that the parameters are $\theta'$ instead of $\theta$. If that is the case, isn't the solution to the maximum likelihood problem simply $\theta'$?

- The authors propose the core idea as a fairly general recipe/meta-algorithm (implicit-MLE), however its specific instantiation via the perturbation based implicit differentiation seems to be an equally important piece of the proposal, and I'm not able to assess which of these is more important even though the presentation and the title suggests that its the former.

- In figure 1, the output needs to be $y$ rather than $L$?


**Time Spent Reviewing:**

3

---

> ### Author Response · Authors · 2021-08-10
> **Answer to Reviewer JsU3**
>
> Thank you for your insightful questions and comments.
>
> > Equation between lines 116 and 117
>
> You are making an excellent point: the target distributions defined in sections 3.1 and 4 should be linked back to the inequality defined between lines 116 and 117. First, as you suggested, we will number the equation. Second, we will integrate the results of Sec 4 as follows.
> For the target distribution $q$ proposed in section 4 and the corresponding loss functions (the combinatorial optimization setting), it is possible to show that the inequality holds. Facts 1 and 2 (and their proofs) show that here the target distribution $q$ for $\tau \to 0$ (which is equivalent to taking the MAP) corresponds precisely to the empirical distribution. In other words, samples from the target distribution $q$ provide the assignments to the discrete random variables which are optimal with respect to the loss at hand. For instance, in the case of the Hamming loss $\ell_H$ (the setting addressed by Fact 1), the target distribution $q$ for $\tau \to 0$  provides the samples $z=y$ for which $\ell_H(z, y) = \ell_H(y, y) = 0$. Therefore, $E_{z \sim q} \ell_{H}(z, y) \leq E_{z \sim p} \ell_{H}(z, y)$ for any p. A similar argument holds for the setting of Fact 2. We will add these statements to Facts 1 and 2 and extend the proofs accordingly.
> Unfortunately, we cannot provide an equally (strong) result for the target distribution of section 3.1. Under the (strong) condition that we have  $E_{z \sim q} \ell(f(z), y) = \ell(f(E_{z \sim q} z), y)$, that is, if we can replace the expectation of the loss by the loss of the expectation of z (the marginals), we are precisely in the setting Domke considered in [1]. Since for $q$ we need to choose a fixed lambda, and we cannot take the limit as in Eq. 9, we have no guarantee that the inequality holds. Nevertheless, Eq. 9 provides a justification that the loss in $q$ is lower than the one in $p$, as for a fixed lambda the gradients of Eq. 9 are equivalent to the maximum likelihood gradients scaled by $1/\lambda$ which I-MLE follows. We’ll clarify this in Section 3.1 and the conclusion.
>
> Thank you again for this important remark.
>
> [1] J. Domke. Implicit differentiation by perturbation. In Advances in Neural Information Processing Systems 23, pages 523–531. 2010.
>
> > In Eq (8), the target $q(z, \theta′)$ definition seems ambiguous as the parameters on the RHS depend on $z$, which is the variable whose distribution is being specified.
>
> Thank you for pointing this out. Indeed, the z on the RHS is sampled and should have a different name -- we will rename it as $\hat{z}$ to avoid any confusion.
>
> > In Eq (12), the target distribution $q$ has the same form as $p$, except that the parameters are $\theta′$ instead of $\theta$: isn't the solution to the maximum likelihood problem simply $\theta′$?
>
> Yes, that is right: however, this fact is not particularly useful in our setting.  In fact, please note that in our setting $\theta=h_v(x)$ is the output of a parameterized function. We are interested in learning the parameters $v$ and do not optimize $\theta$ directly.
> One may think of using $\theta - \theta’$ as the gradients, but this would not be equivalent to the maximum likelihood gradients which are, instead, the difference between the marginals of the two distributions. We showed in Facts 1 and 2 that our method is producing precisely those MLE  gradients in the settings described in Section 4. In fact, one can show that $\theta - \theta’$ corresponds to a scaled version of the gradients of the straight-through estimator (STE) which, as discussed in lines 155-160, does not take the constraints (modelling the dependencies between the variables) into account and which we have shown to underperform empirically.
>
> > The core idea as a fairly general recipe/meta-algorithm (Implicit-MLE); its specific instantiation via the perturbation based implicit differentiation seems to be an equally important piece of the proposal.
>
> Thank you for the comment. In the updated version, we will add a paragraph to the introduction where we outline the specific contributions. Both are important ingredients of the proposed approach: I-MLE describes a general framework for which one needs to define a target distribution q, and noise-based perturbations are used for computing the gradient. Both contributions are novel and crucial for the success of our approach.
>
> > In Figure 1, the output needs to be y rather than L.
>
> Yes, thank you. We will add y to clarify the figure.
>
> We appreciate your helpful questions. Your feedback will help to improve the paper significantly.

---

> > ### Comment · Reviewer_JsU3 · 2021-08-29
> > **reply**
> >
> > Thanks to the authors for the response and acknowledging some of the concerns in the original review -- I've edited my review score to a marginal accept conditional on a successful presentation of these clarifications in a revised paper.

---

### Official Review · Reviewer_MvPH · 2021-07-26

**Rating:** 6
**Confidence:** 4

**Summary:**

The paper proposes I-MLE, a class of gradient estimation techniques for computational graphs involving sampling from discrete exponential family distributions. The techniques are built on top of the perturb-and-MAP strategies for sampling from such distributions, extended to account for more general noise perturbation strategies via Sum of Gamma distributions in the forward pass and (biased) gradient estimators that combine straight-through-estimation with perturbation-based differentiation in the backward pass. Finally, the paper provides empirical evaluation on real and synthetic benchmarks that indicates strong and competitive performance of the proposed approach.

**Limitations And Societal Impact:**

Yes.

**Main Review:**


- In an attempt to optimize computation graphs with discrete latent variables, the paper revisits the classic MAP-and-perturb scheme for sampling via optimization. While these strategies are known, their use and extension in an end-to-end pipeline is novel to this work. Moreover, this scheme presents an interesting departure from the predominant strategies: score function estimation with control variates and continuous relaxations+reparameterization.

  While I am generally in favor of fresh takes, my general worry is that in doing so, the paper adds more complexity to both the forward and backward pass. In the forward pass, there is a need for a (black-box) MAP solver and the backward pass relies on a new biased estimate of the gradient. There are also additional hyperparameters, such as lambda or the choice of q more generally, that are not needed for alternative baseline gradient estimators. Having said that, the authors are able to demonstrate good performance, thereby lending credibility that the extra machinery is perhaps worthy of further exploration.

- The writing is mostly clear, except for a key piece of intuition:
In L119-120, the authors say that "The intuition is that making p more similar to q will also reduce the loss" --- why is that the case? does it hold universally for all q's that satisfy the equation following L116?

- The results in Tables 1 and 2 use different q's (eq. 8 and 12). What would the performance be if the q's for Tables 1 and 2 were switched?

- I would also move Algorithm 1 to the main paper. It summarizes the algorithm well enough to aid clarity.

- The estimator name, I-MLE, is somewhat too general. The connections with MLE hold only for very specific choice of q's and loss functions (sec 4). Or is there a broader connection that holds more generally?

- Could the SoG noise distribution also improve the default Gumbel-softmax relaxation? In general, it could be insightful to report results from a SoG-softmax relaxation to assess the general-purpose utility of this noise distribution.

- Empirically, how does this method scale with the number of categories for the discrete VAE?

- There are some missing related works:

Kim, C., Sabharwal, A. and Ermon, S.Exact sampling with integer linear programs and random perturbations. AAAI, 2016 -- Guarantees for MAP and perturb based methods

Grover, A., Wang E., Zweig A., and Ermon, S. "Stochastic optimization of sorting networks via continuous relaxations.", ICLR 2019  -- Continuous relaxation for combinatorial discrete distributions


**Time Spent Reviewing:**

8

---

> ### Author Response · Authors · 2021-08-10
> **Answer to Reviewer MvPH**
>
> Thank you for your insightful questions and comments.
>
>
> > Complexity in both the forward and backward pass: in the forward pass, there is a need for a (black-box) MAP solver and the backward pass relies on a new biased estimate of the gradient.
>
> Indeed, the proposed method relies on the ability of computing MAP states for a given distribution. However, there are several cases (e.g. when using black-box combinatorial solvers, such as a shortest path algorithm) where computing marginals and sampling is not tractable but  MAP inference is.
> Moreover, in the Appendix, Section D, we empirically compare the complexity of perturb-and-MAP with a Sum-of-Gamma noise distribution, with faithful sampling and expectations, showing that the proposed method has significantly better sampling properties.
>
>
> > In L119-120, the authors say that "The intuition is that making $p$ more similar to $q$ will also reduce the loss" --- why is that the case? does it hold universally for all $q$'s that satisfy the equation following L116?
>
> If we assume that $q$ incurs a lower loss than p and has the same set of constraints, then `` moving $p$ toward $q$ '' (reducing the KL divergence between $p$ and $q$) asymptotically reduces the expectation of the loss. This is due to the consistency property of the maximum likelihood estimator which guarantees that asymptotically the parameters of p converge in probability to the parameters of $q$. As another reviewer suggested, we will number the equation between lines 116-117 and better link the text to it. Please also see the answer to the related first question of reviewer JsU3.
>
>
> > Results in Tables 1 and 2 use different q's: what would the performance be if the q's for Tables 1 and 2 were switched?
>
> As you rightfully point out (and as mentioned in the paper), the choice of the target distribution is a hyperparameter. We propose in sections 3.1 and 4 two widely applicable classes of target distributions. The experiments of Table 2 pertain to a setting in which the third component of the model ($f_u$) is trivial and one has access to direct supervision regarding the discrete distribution. In such cases, we show that the target distribution introduced in Eq. (12) of Section 4 is a good candidate distribution (see Facts 1 and 2) as its use is equivalent to performing maximum-likelihood learning with respect to the empirical distribution.
> This is no longer the case for the experiments of Table 1, where $f_u$ is nontrivial and the supervision for $p$ is only implicit. In this scenario, we do not expect Eq. (12) to work properly and we observed this in preliminary experiments. We will clarify this in the text.
> On the other hand, one may instead use the target distribution of Sec 3.1 for the experiments of Table 2. Results for this are shown in the third column of Table 2 (see also footnote 10). Yet, we empirically found that the distribution of Section 4 works better in this setting.
>
>
> > Move Algorithm 1 to the main paper.
>
> Thank you -- we will do this.
>
>
> > Could the SoG noise distribution also improve the default Gumbel-softmax relaxation?
>
> The SoG noise distribution reduces to a Gumbel distribution when $k=1$; i.e. $\text{SoG}(1, \tau) = \text{Gumbel}(0, \tau)$. We also remark on this in lines 199-205 in the submission.
>
>
> > How does this method scale with the number of categories for the discrete VAE?
>
> Assuming by categories you mean the dimensionality of the latent code (denoted $m$ in the paper), the answer largely depends on the specific choice of $p$ and the complexity of computing MAP states. If $p$ is the $k$-subset distribution (the distribution used in the experiments), then the method scales linearly in $m$ as computing MAP states of such distribution has an $O(m)$ time complexity.  We would like to also refer the reviewer to Figure 7 in the appendix for a comparison of runtimes of perturb-and-MAP sampling vs faithful sampling and full expectation for the $k$-subset distribution.
>
>
> > Missing references
>
> Thanks for pointing these out, we will include them in the final version.
>
>
> Thanks again for your time. Please let us know if we misunderstood any of your points or questions.

---

> > ### Comment · Reviewer_MvPH · 2021-08-25
> > **Reply**
> >
> > Thanks for your response and addressing some of my concerns. Tables {1,2} and the SoG noise distribution could have been better addressed with experimental validation.

---

> > > ### Author Response · Authors · 2021-08-29
> > > **Additional experiments**
> > >
> > >
> > > Thank you for pointing this out. As stated in our previous answer, experiments of Table 2 use the target distribution introduced in Eq. (12). As additional experiments, we also consider the setting where the target distribution is the one given in Eq. (8), and where noise samples were drawn from a SoG distribution.
> > >
> > > Results are summarised in the following table, and show that Eq. (8) and Eq. (12) yield very similar results for $\tau = 0.01$. For Eq (8), results seem to degrate for larger temperatures $\tau$. We conducted extensive experiments for multiple values of $\tau$ and Warcraft map sizes $k$, and we will report them in the revised version of the paper.
> > >
> > > | Model | Results for k=30 (Mean Accuracy $\pm$ Std. Dev.) |
> > > | --- | --- |
> > > | I-MLE ($\mathtt{MAP}-\mathtt{MAP}$) -- Eq. (12) | 93.7 $\pm$ 0.6 |
> > > | I-MLE ($\mu-\mu$) -- Eq. (12) | 93.6 $\pm$ 0.4 |
> > > I-MLE ($\lambda = 20$) -- Eq. (8) | 93.6 $\pm$ 0.5 |
> > > I-MLE ($\lambda = 10$, $\tau = 1$) -- Eq. (8) | 79.8 $\pm$ 0.3 |
> > > I-MLE ($\lambda = 20$, $\tau = 0.01$) -- Eq. (8) | 93.8 $\pm$ 0.3 |
> > >
> > > Here are the results for multiple values of $k$:
> > >
> > > | Model | $k$ | Results (Mean Accuracy $\pm$ Std. Dev.) |
> > > | --- | --- | --- |
> > > I-MLE ($\mathtt{MAP}-\mathtt{MAP}$) -- Eq. (12) | 12 | 95.2 $\pm$ 0.3|
> > > I-MLE ($\mathtt{MAP}-\mathtt{MAP}$) -- Eq. (12) | 18 | 94.4 $\pm$ 0.5|
> > > I-MLE ($\mathtt{MAP}-\mathtt{MAP}$) -- Eq. (12) | 24 | 93.2 $\pm$ 0.2|
> > > I-MLE ($\mathtt{MAP}-\mathtt{MAP}$) -- Eq. (12) | 30 | 93.7 $\pm$ 0.6|
> > > I-MLE ($\mu-\mu$) -- Eq. (12) | 12 | 97.2 $\pm$ 0.5 |
> > > I-MLE ($\mu-\mu$) -- Eq. (12) | 18, | 95.8 $\pm$ 0.7|
> > > I-MLE ($\mu-\mu$) -- Eq. (12) | 24, | 94.3 $\pm$ 1.0|
> > > I-MLE ($\mu-\mu$) -- Eq. (12) | 30, | 93.6 $\pm$ 0.4|
> > > I-MLE ($\lambda = 20$) -- Eq. (8) | 12, | 95.2 $\pm$ 0.7|
> > > I-MLE ($\lambda = 20$) -- Eq. (8) | 18, | 94.7 $\pm$ 0.4|
> > > I-MLE ($\lambda = 20$) -- Eq. (8) | 24 | 93.8 $\pm$ 0.3|
> > > I-MLE ($\lambda = 20$) -- Eq. (8) | 30 | 93.6 $\pm$ 0.5|
> > > I-MLE ($\lambda = 10$, $\tau = 1$) -- Eq. (8) | 12 | 85.9 $\pm$ 0.3|
> > > I-MLE ($\lambda = 10$, $\tau = 1$) -- Eq. (8) | 18 | 84.0 $\pm$ 0.5|
> > > I-MLE ($\lambda = 10$, $\tau = 1$) -- Eq. (8) | 24 | 79.7 $\pm$ 0.6|
> > > I-MLE ($\lambda = 10$, $\tau = 1$) -- Eq. (8) | 30 | 79.8 $\pm$ 0.3|
> > > I-MLE ($\lambda = 20$, $\tau = 0.01$) -- Eq. (8) | 12 | 95.1 $\pm$ 0.4|
> > > I-MLE ($\lambda = 20$, $\tau = 0.01$) -- Eq. (8) | 18 | 94.4 $\pm$ 0.4|
> > > I-MLE ($\lambda = 20$, $\tau = 0.01$) -- Eq. (8) | 24 | 93.7 $\pm$ 0.4|
> > > I-MLE ($\lambda = 20$, $\tau = 0.01$) -- Eq. (8) | 30 | 93.8 $\pm$ 0.3|
> > >
> > > We reiterate that for the setting of Table 1, preliminary experiments showed that the target distribution of Eq (4) did not yield good results. This is to be expected since that target distribution is meaningful in the context described in Sec. 4 (i.e. where there are forms of explicit supervision over the discrete states).
> > >
> > > We hope this addresses the reviewer’s concern, and kindly ask whether the reviewer has any other specific experimental setting in mind.

---

### Decision · Program_Chairs · 2021-09-27

**Decision:**

Accept (Poster)

**Comment:**

The paper presents new methods for gradient estimation in computation graphs with certain types of discrete stochastic variables. Reviewers all voted accept (scores 6, 6, 6, 8). The consensus was that the paper has fresh ideas and there is evidence in favor of the method working well, but it also has some tradeoffs, e.g., in terms of added complexity in the computations. Several reviewers raised (non-fatal) concerns about clarity, which the authors are encouraged to address in the final revision. The meta-reviewer recommends accept.